**JCB** Journal of Cell Biology

# Pex30-like proteins function as adaptors at distinct ER membrane contact sites

Joana Veríssimo Ferreira and Pedro Carvalho

**Membrane lipids and proteins synthesized in the ER are used for de novo assembly of organelles, such as lipid droplets and peroxisomes. After assembly, the growth of these organelles is supported by ER-derived lipids transferred at membrane contact sites (MCSs). How ER sites for organelle biogenesis and lipid transfer are established and regulated is unclear. Here, we investigate how the ER membrane protein Pex30 and its family members Pex28, Pex29, Pex31, and Pex32 target and function at multiple MCSs. We show that different Pex30 complexes function at distinct ER domains and MCSs. Pex30 targets ER–peroxisome MCSs when bound to Pex28 and Pex32, organizes the nuclear–vacuolar junction when bound to Pex29, and promotes the biogenesis of lipid droplets independently of other family members. Importantly, the reticulon homology domain (RHD) mediates the assembly of the various Pex30 complexes. Given the role of RHD in membrane shaping, our findings offer a mechanistic link between MCS and regulation of membrane curvature.**

## Introduction

The ER is the largest membrane-bound organelle and the main site for the biosynthesis of membrane proteins and lipids. The majority of these molecules traffic to other cellular destinations. Membrane contact sites (MCSs), where membranes of distinct organelles are brought into close proximity, have emerged as a major lipid trafficking route (Scorrano et al., 2019; Wu et al., 2018). For example, the ER establishes conspicuous MCSs with all other cellular organelles, including lipid droplets (LDs), peroxisomes, and vacuoles/lysosomes (Valm et al., 2017; Shai et al., 2018). In these cases, MCSs have been implicated in facilitating the transfer of ER-synthesized lipids to support the growth of associated organelles (Wu et al., 2018; Reinisch and Prinz, 2021; Prinz et al., 2020). These MCSs also control the spatial distribution of organelles and ultimately affect their inheritance into the daughter cells during cell division (Knoblach and Rachubinski, 2019). Despite their importance in cell homeostasis, the molecules mediating MCSs and their regulation remain poorly understood.

The ER also serves as a platform for de novo assembly of other organelles, namely LDs and preperoxisomal vesicles (PPVs), which mature into peroxisomes (Joshi et al., 2016). While both LDs and peroxisomes are important in cellular lipid metabolism, they are structurally and functionally distinct (Kohlwein et al., 2013). LDs are storage organelles with a core filled with neutral lipids enclosed by a phospholipid monolayer. In contrast, yeast peroxisomes are delimited by a phospholipid bilayer defining a protein-rich matrix primarily involved in fatty acid β-oxidation.

Despite the functional and structural differences between LDs and peroxisomes, recent studies in yeast showed that the ER domains for the biogenesis of both organelles coincide (Wang et al., 2018; Joshi et al., 2018). These discrete sites for LD and peroxisome biogenesis are marked by the ER integral membrane protein Pex30. Newly formed organelles often remain in close contact with the ER at the Pex30 marked sites. Pex30 accumulation at ER sites for organelle biogenesis is particularly prominent in cells lacking the seipin Sei1 or Ldb16, components of the seipin complex, which facilitate LD biogenesis. In fact, simultaneous deletion of seipin and Pex30 strongly compromises the biogenesis of both LDs and peroxisomes. These double-mutant cells accumulate aberrant levels of both phospholipids and neutral lipids in the ER and display a slow growth phenotype (Wang et al., 2018).

During peroxisome biogenesis, Pex30 appears to promote local high membrane curvature required for the budding of PPVs. Pex30 membrane tubulation activity depends on its reticulon homology domain (RHD), as shown by in vitro experiments (Joshi et al., 2016). Whether Pex30 acts in a similar fashion during LD biogenesis is unknown.

Pex30 is the founding member of a family of ER membrane protein—the Pex30 family—that also includes Pex28, Pex29, Pex31, and Pex32. Members of this family are characterized by a similar domain architecture consisting of an RHD, mediating ER membrane association, and a C-terminal Dysferlin (DysF) domain of unknown function (Fig. 1 A). Early genetic studies

---

Sir William Dunn School of Pathology, University of Oxford, Oxford, UK.

Correspondence to Pedro Carvalho: pedro.carvalho@path.ox.ac.uk.



showed that mutations in any of the Pex30 family members resulted in aberrant peroxisome numbers and morphology, hinting at similar albeit nonredundant functions among the family members (Vizeacoumar et al., 2004, 2003; David et al., 2013; Mast et al., 2016). Peroxisome morphology defects were also observed in other yeast species with mutations in Pex30-like proteins (Brown et al., 2000; Tam and Rachubinski, 2002; Yan et al., 2008; Wu et al., 2020). Loss of Pex30 also affects the distribution and dynamics of peroxisomes, affecting their inheritance by daughter cells during mitosis (Knoblach et al., 2013; David et al., 2013). Whether these peroxisome defects arise from defective biogenesis (Joshi et al., 2016; Mast et al., 2016), aberrant ER–peroxisome MCSs (David et al., 2013; Mast et al., 2016), or both is unclear. However, recent studies in *Hansenula polymorpha* showed that mutations in Pex28 and Pex32 homologues were rescued by an artificial ER–peroxisome tether, suggesting that the defects arise from defective MCSs between these organelles (Wu et al., 2020). Moreover, proteomics-based approaches showed interactions between Pex30 family members, suggesting that they function as a biochemical complex (Mast et al., 2016; David et al., 2013). However, these potential relationships between Pex30 family members have not been explored, and how the various members of the family cooperate in maintaining organelle homeostasis is unknown.

Here, we investigate the relationship between the Pex30 family members. We show that Pex31 functions independently of all the other family members. While Pex30 also appears to function by itself during LD biogenesis, at other MCSs, the function of Pex30 depends on the assembly of mutually exclusive complexes with Pex29 and Pex28/Pex32. The complex containing Pex30/Pex28/Pex32 targets ER–peroxisome MCSs, while the Pex30/Pex29 complex concentrates mainly at a distinct MCS, the nuclear–vacuolar junction (NVJ), where it regulates a population of LDs. Finally, we show that both the RHD and the DysF domain play essential but distinct roles in Pex30 function.

## Results

### Pex30 functions independently of its family members in LD biogenesis

Pex30 was recently implicated in the biogenesis of LDs (Choudhary et al., 2020; Joshi et al., 2018; Wang et al., 2018). In particular, efficient LD formation in *sei1Δ* and *ldb16Δ* seipin complex mutants requires Pex30. Inhibition of LD biogenesis in double mutants lacking simultaneously Sei1 and Pex30 was accompanied by a strong growth defect. In contrast, deletion of any other Pex30 family member in seipin mutants resulted in normal cell growth (Wang et al., 2018). These results suggest that Pex30 has a unique role in LD formation that is not shared by the other Pex30 family members Pex28, Pex29, Pex31, and Pex32, with sequence similarity (Fig. S1 A) and the same domain architecture (Fig. 1 A). To further characterize this unique role of Pex30, we analyzed LDs in double mutants lacking *sei1Δ* and individual members of the Pex30 family. In *sei1Δ* mutant, we observed few LDs per cell or a single aggregate consisting of tiny LDs (Fig. 1 B), as expected (Fei et al., 2008; Szymanski

et al., 2007; Wang et al., 2014; Grippa et al., 2015). In *sei1Δpex30Δ* cells, a few aberrant LDs were observed, but most neutral lipids were dispersed throughout the ER membrane, originating a unique dispersed BODIPY stain (Fig. 1 B), as described earlier (Wang et al., 2018; Joshi et al., 2018). In contrast, combined deletion of Sei1 with Pex28, Pex29, Pex31, or Pex32 did not show dispersed structures labeled by BODIPY or a block in LD formation (Fig. 1 B). In fact, LD formation was not inhibited even in *sei1Δpex28Δpex29Δpex31Δpex32Δ* quintuple mutant (Fig. 1 B). Consistently, this mutant displayed comparable levels and distribution of Pex30 (Fig. S1, B and C). Thus, under some conditions, such as *sei1Δ* cells, Pex30 functions independently of its family members to promote LD budding.

Despite the similar domain organization among the Pex30 family members (Fig. 1 A), the presence of the C-terminal Domain of Unknown Function 4196 (DUF4196) is exclusive to Pex30. To test the role of this domain in Pex30-dependent LD formation, we deleted the DUF4196 region in endogenous Pex30 to generate Pex30$^{DUF\Delta}$. The localization and the levels of Pex30$^{DUF\Delta}$ appeared comparable to those of full-length Pex30 in both WT cells and seipin mutants (Fig. 1 C). Importantly, LD formation was not blocked in *sei1ΔPEX30$^{DUF\Delta}$* cells, indicating that DUF4196 is dispensable for Pex30 essential function in seipin mutants (Fig. 1 C).

The unique requirement of Pex30 in seipin mutants could potentially be explained by higher expression levels of Pex30 relative to the other family members. To test this possibility, we analyzed the relative levels of endogenously HA-tagged Pex30 family members. Pex30 is indeed the most highly expressed family member, followed by Pex29 and Pex31. In contrast, Pex28 and Pex32 were expressed at much lower levels (Fig. 1, D and E). Next, we tested whether overexpression of the various family members could compensate for the loss of Pex30. However, neither the LD defects with the appearance of unique dispersed BODIPY structures (Fig. 1 F) nor the growth defects (Fig. S1, D and E) of *sei1Δpex30Δ* cells were reverted by overexpression of Pex28, Pex29, or Pex31. Altogether, these data show that Pex30 has unique and independent functions among its family members.

### Pex30 binds to and stabilizes Pex28, Pex29, and Pex32 but not Pex31

Our data showing that Pex30 can function independently of other family members contrast with earlier work indicating that it interacts with other members of the Pex30 family (David et al., 2013; Mast et al., 2016). Therefore, we reevaluated the relationship between the Pex30 family members. We observed that endogenously HA-tagged Pex28, Pex29, and Pex32 coprecipitated with Pex30. In contrast, its closest relative, Pex31, did not interact with Pex30 (Fig. 2 A). Consistent with the immunoprecipitation result, we observed that in the absence of Pex30, the levels of its binding partners Pex28, Pex29, and Pex32 dropped dramatically (Fig. 2 B). On the other hand, the steady-state levels of Pex31 were insensitive to the presence of Pex30, in agreement with the lack of interaction between the two proteins. Moreover, the effect of Pex30 on the levels of Pex28, Pex29, and Pex32 was specific, as the levels of these proteins

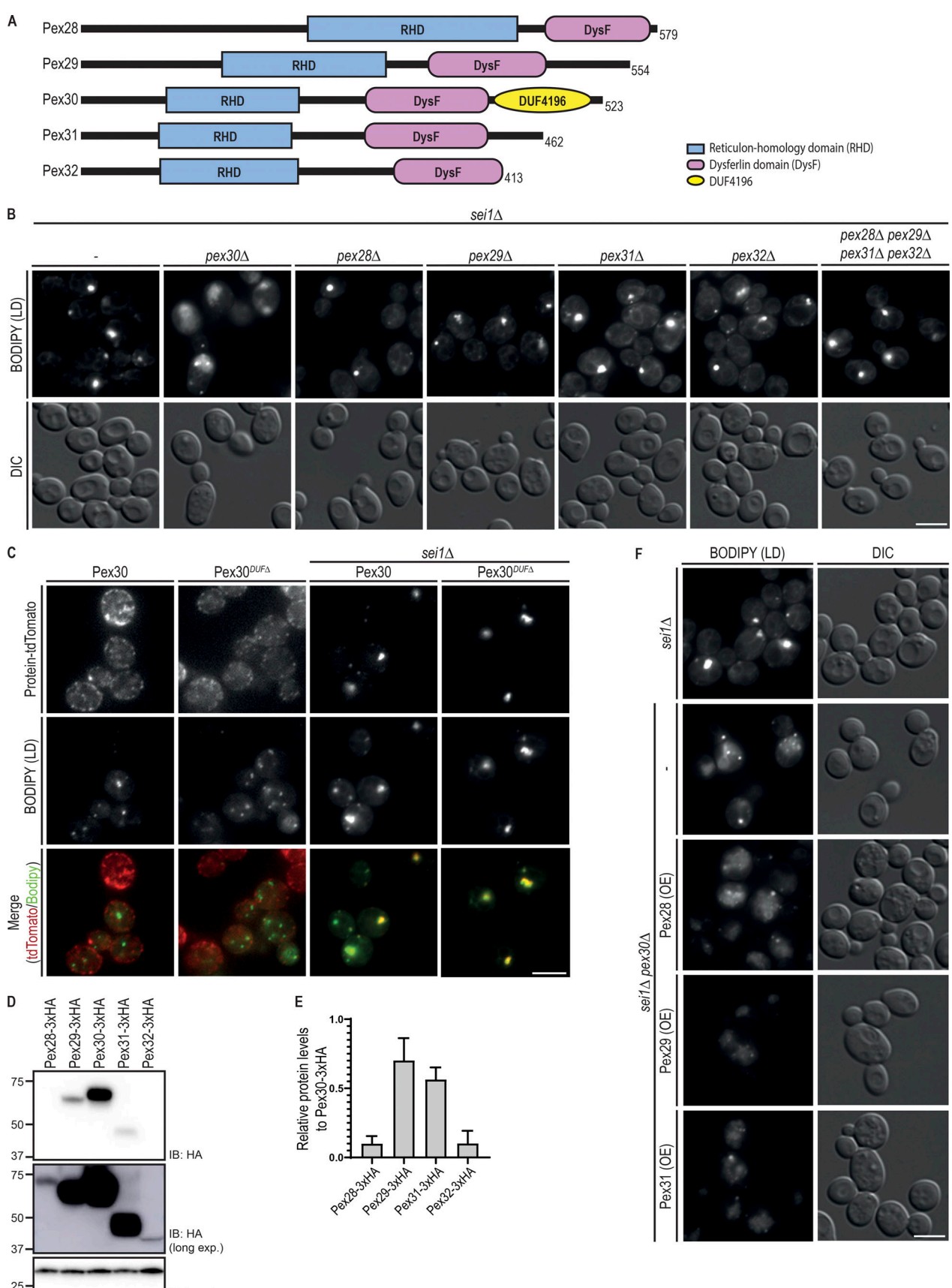

Figure 1. **Pex30 promotes LD biogenesis independently of its family members. (A)** Domain organization of Pex30 family of proteins. RHD and DysF domains were identified using the structure-based prediction software HHpred (Hildebrand et al., 2009). Pex30 also has an exclusive domain, the DUF4196.

**(B)** Analysis of LDs in cells with the indicated genotype. LDs were stained with the neutral lipid dye BODIPY 493/503. Please note that dispersed BODIPY structures were detected exclusively in *sei1Δpex30Δ* cells. Bar, 5 μm. DIC, differential interference contrast. **(C)** Localization of Pex30 and Pex30$^{DUFΔ}$, a Pex30 C-terminal truncation lacking DUF4196, in WT and *sei1Δ* cells. Both Pex30 and Pex30$^{DUFΔ}$ were expressed from the endogenous Pex30 locus as a fusion to tdTomato fluorescent protein. LDs were stained with the neutral lipid dye BODIPY 493/503. Bar, 5 μm. **(D)** Steady-state levels of Pex30 and its family members. Pex30 family proteins were expressed from the endogenous locus as a C-terminal HA fusion. Whole-cell extracts were prepared from exponentially growing cells, separated by SDS-PAGE, and analyzed by Western blotting. Pex30 family members were detected with anti-HA, and Dpm1, used as loading control, was detected with anti-Dpm1 antibodies. Due to the low levels of Pex28 and Pex32, a longer exposure was included to confirm and quantify their expression. IB, immunoblot. **(E)** Relative protein levels of Pex30 family members. The levels of Pex30 family proteins in relation to Pex30 determined as in (D). The average of three independent experiments is shown. Bars represent SD. **(F)** Analysis of LDs in cells with the indicated genotype. LDs were stained as in B. Please note that dispersed BODIPY structures typical from *sei1Δpex30Δ* cells were also observed in cells overexpressing Pex28, Pex29, or Pex31. Bar, 5 μm.

were unaffected in *pex31Δ* cells (Fig. 2 B). Thus, Pex30 binds to and stabilizes Pex28, Pex29, and Pex32. In contrast, Pex31 appears to function independently of the other family members.

### Pex28/Pex32 and Pex29 form mutually exclusive complexes with Pex30

Next, we asked whether Pex30 and its binding partners Pex28, Pex29, and Pex32 were all present in a single biochemical complex or instead formed multiple independent complexes. Endogenous Pex28, Pex29, and Pex32 were simultaneously expressed as MYC, V5, and HA fusion proteins, respectively, and cell extracts were subjected to immunoprecipitation with antibodies to the various epitope tags. We observed that Pex28-MYC precipitates contained Pex30 and Pex32 but not Pex29 (Fig. 2 C). Similarly, Pex32-HA immunoprecipitated Pex28 and Pex30 but not Pex29. Conversely, Pex29-V5 interacted with Pex30 but not with Pex28 or Pex32 (Fig. 2 C). Thus, Pex30 assembles mutually exclusive complexes with Pex29 and Pex28/Pex32.

Interestingly, cells lacking Pex29, the most abundant of Pex30 partners, have about twofold higher levels of Pex32 (Fig. S2, A and B), suggesting increased Pex30 availability in *pex29Δ* mutants. This is consistent with a model in which Pex28/Pex32 and Pex29 compete for Pex30 binding. While more work will be required to dissect the organization of these complexes, in *pex32Δ* cells, Pex30 still interacts with Pex28 even if this protein is present at much lower levels (Fig. S2 C). Conversely, in *pex28Δ* cells, Pex30 does not interact with Pex32. These data indicate that Pex28 mediates the interaction between Pex30 and Pex32.

### Pex30/28/32 and Pex30/29 complexes target different MCSs

While many ER membrane proteins distribute homogeneously throughout the ER, Pex30 localizes to discrete ER domains (Fig. 1 C). These were shown to correspond to ER contacts with peroxisomes and LDs as well as biogenesis sites for these organelles (Mast et al., 2016; David et al., 2013; Joshi et al., 2016, 2018; Wang et al., 2018; Choudhary et al., 2020). We then asked whether Pex30-binding partners would localize preferentially to specific ER regions proximal to LDs and peroxisomes. Endogenous Pex28, Pex29, and Pex32 were expressed as a fusion to monomeric NeonGreen (mNG) in cells where both peroxisomes and LDs were labeled with mCherry-PTS1 and monodansyl pentane (MDH), respectively. Like Pex30, both Pex28 and Pex32 localized as discrete foci. However, these were much sparser, consistent with lower levels of these proteins (Fig. 3 A). Importantly, all peroxisomes labeled by mCherry-PTS1 were apposed to Pex28 and Pex32 foci, but these proteins were never

in proximity to LDs (Fig. 3, A–C). Thus, the Pex30/28/32 complex targets ER–peroxisome contacts. In exponentially growing cells, Pex29 appeared as faint foci densely distributed throughout the peripheral ER. While it was difficult to assign its localization to specific contact sites, we observed that a fraction of both peroxisomes (37% ± 4%, Fig. 3 B) and LDs (47% ± 4%, Fig. 3 C) were proximal to Pex29 foci. Curiously, a few cells showed bright Pex29 structures (Fig. 3 A). These brighter structures were elongated and more centrally located within the cell. These features were reminiscent of the NVJ, an MCS between the nuclear ER and the vacuole, the yeast equivalent to the lysosome (Pan et al., 2000). The NVJ expands in cells under conditions of metabolic stress, such as glucose depletion. For example, as cells exit exponential growth after glucose exhaustion and rewire their metabolism for stationary phase, a process known as diauxic shift, the NVJ dramatically expands and serves as a platform for LD biogenesis (Hariri et al., 2018). We therefore analyzed the localization of Pex29 in early stationary-phase cells, which display a prominent NVJ as detected by the marker protein Nvj1 (Fig. 3 D). The distributions of Nvj1 and these Pex29 structures showed a complete overlap, confirming Pex29 localization to the NVJ. Importantly, under the same growth conditions, Pex30 also labeled the NVJ (Fig. 3 D), indicating that this MCS is targeted by the Pex30/Pex29 complex. In contrast, Pex32 was never observed at the NVJ (Fig. 3 D), while Pex28 levels become undetectable in stationary phase (Fig. S3). Thus, consistent with our biochemical analysis, Pex30/Pex28/Pex32 and Pex30/Pex29 complexes localize to distinct MCS, and their targeting appears regulated by nutrient availability.

### Spatial organization of LDs by the Pex30/Pex29 complex

The roles of Pex30 at ER–peroxisome contacts have been characterized (Vizeacoumar et al., 2004; Mast et al., 2016; David et al., 2013; Joshi et al., 2016). In agreement with those studies, we observed that mutations in Pex30 or its family members result in increased numbers of peroxisomes (Fig. S4 A). However, there is no information on how Pex30, together with Pex29, targets and functions at the NVJ, whose expansion in stationary phase supports LD formation (Hariri et al., 2018). To gain insight on these issues, we asked whether Pex29 was necessary for Pex30 localization to the NVJ. We observed that in *pex29Δ* mutants, the brighter Pex30-mNG structures characteristic of accumulation at the NVJ were absent, suggesting that Pex30 failed to localize to the NVJ (Fig. 4 A). Importantly, both the overall Pex30 levels and their localization to other ER domains were unaffected (Figs. 4 A and S1 C). In addition, Pex30 localization was unaffected in *pex28Δ* and

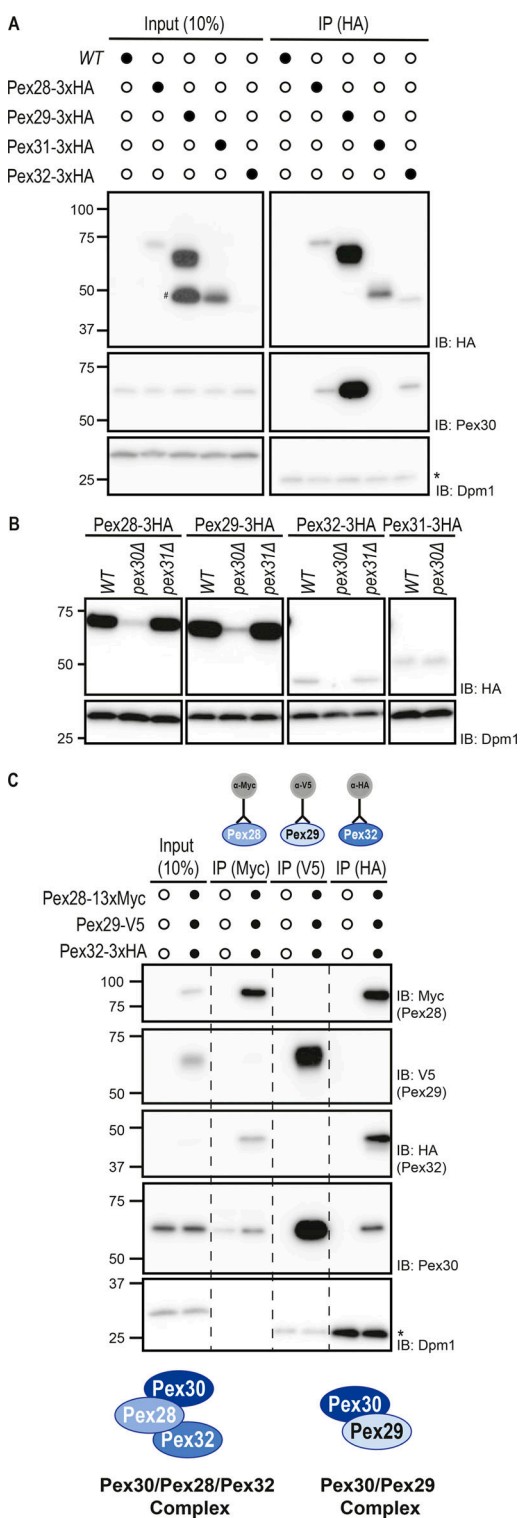

**Figure 2. Pex30 binds to and stabilizes Pex28, Pex29, and Pex32 into two distinct complexes. (A)** Pex30 coprecipitates with Pex28, Pex29, and Pex32 but not Pex31. Crude membranes of cells expressing the indicated HA-tagged proteins were solubilized, detergent extracts were immunoprecipitated (IP), and eluted proteins were analyzed by SDS-PAGE and Western blotting. #, Pex29 degradation product; *, IgG light chain. IB, immunoblot. **(B)** Steady-state levels of endogenous HA-tagged Pex28, Pex29, Pex31, and Pex32 in WT, *pex30Δ*, and *pex31Δ* cells, during stationary phase. Whole-cell extracts were analyzed as in Fig. 1 D. **(C)** Pex30 forms mutually exclusive complexes with Pex28/Pex32 and Pex29. Crude membrane fractions of cells expressing

*pex32Δ* mutants (Fig. 4 A). Thus, Pex29 appears to have a specific role in directing Pex30 to the NVJ.

To further characterize Pex30 localization to the NVJ, we performed time-course experiments to follow the transition from exponential growth through diauxic shift and to stationary phase, during which the NVJ assembles and expands (Hariri et al., 2018). Formation of the NVJ was monitored by expression of endogenous Nvj1 fused to tandem dimer Tomato (Nvj1-tdTomato). In WT cells, Nvj1-tdTomato levels were undetectable during exponential growth but prominently labeled the NVJ as cells transitioned to diauxic shift (Fig. 4 B). Expansion of the NVJ also led to the accumulation of adjacent LDs (Fig. 4 C), as expected (Hariri et al., 2018). During this period, we also observed an increase of Pex30-mNG at the NVJ (Fig. 4 B). As in WT cells, *pex29Δ* mutant showed an increase in Nvj1-tdTomato levels during diauxic shift. However, most cells displayed abnormal Nvj1-tdTomato localization that appeared distributed throughout the nuclear ER (Fig. 4, B and D). Aberrant Nvj1-tdTomato localization in *pex29Δ* mutants was observed even at later time points; in a fraction of cells, however, Nvj1-tdTomato eventually concentrated at the NVJ (Fig. 4, B and D). Increased Nvj1 localization to the NVJ at later time points was accompanied by a concomitant accumulation of adjacent LDs (Fig. 4 C). Importantly, irrespective of Nvj1 localization, Pex30 failed to localize to the NVJ in *pex29Δ* cells. These defects were not due to growth conditions, as they were observed in cells grown in minimal medium (Fig. S4, B and C). Moreover, they were specific to Pex30/Pex29 complex, as LDs clustering at the NVJ were unaffected in *pex28Δ* and *pex32Δ* mutants (Fig. 4 E). Together, our data indicate that timely Nvj1 localization and optimal LD clustering to the NVJ require the Pex30/Pex29 complex.

**Distinct but essential roles of Pex30 RHD and DysF domains**
To determine the Pex30 regions involved in the interactions with its family members, we generated domain deletion mutants (Fig. S5 A). We reasoned that mutations affecting Pex30 binding would result in destabilization of its partners, as observed in *pex30Δ* cells (Fig. 2 B). Therefore, we analyzed the levels of endogenous Pex28, Pex29, and Pex32 tagged with MYC, V5, and HA, respectively. Partial deletions of Pex30 RHD resulted in lower steady-state levels of Pex30 as well as its partners Pex28, Pex29, and Pex32 (Fig. S5 B). In contrast, Pex28, Pex29, and Pex32 levels were unchanged in cells expressing Pex30 derivatives lacking the DysF or the DUF4196 domains (Fig. S5 B). To test whether Pex30 RHD is sufficient to bind and stabilize its partners, we generated Pex30(60–283), a Pex30 mutant lacking both the N-terminal region and the DysF and DUF4196 domains (Fig. 5 A). Remarkably, when expressed from the endogenous PEX30 locus, Pex30(60–283) bound (Fig. S5 C) and stabilized (Fig. 5 B) Pex28, Pex29, and Pex32. These data suggest that the

endogenous Pex28, Pex29, and Pex32 fused, respectively, to MYC, V5, and HA epitope tags, or untagged proteins as control were detergent solubilized, and extracts were subjected to immunoprecipitation with anti-MYC, V5, or HA antibodies. Eluted proteins were analyzed by Western blotting. *, IgG light chain.

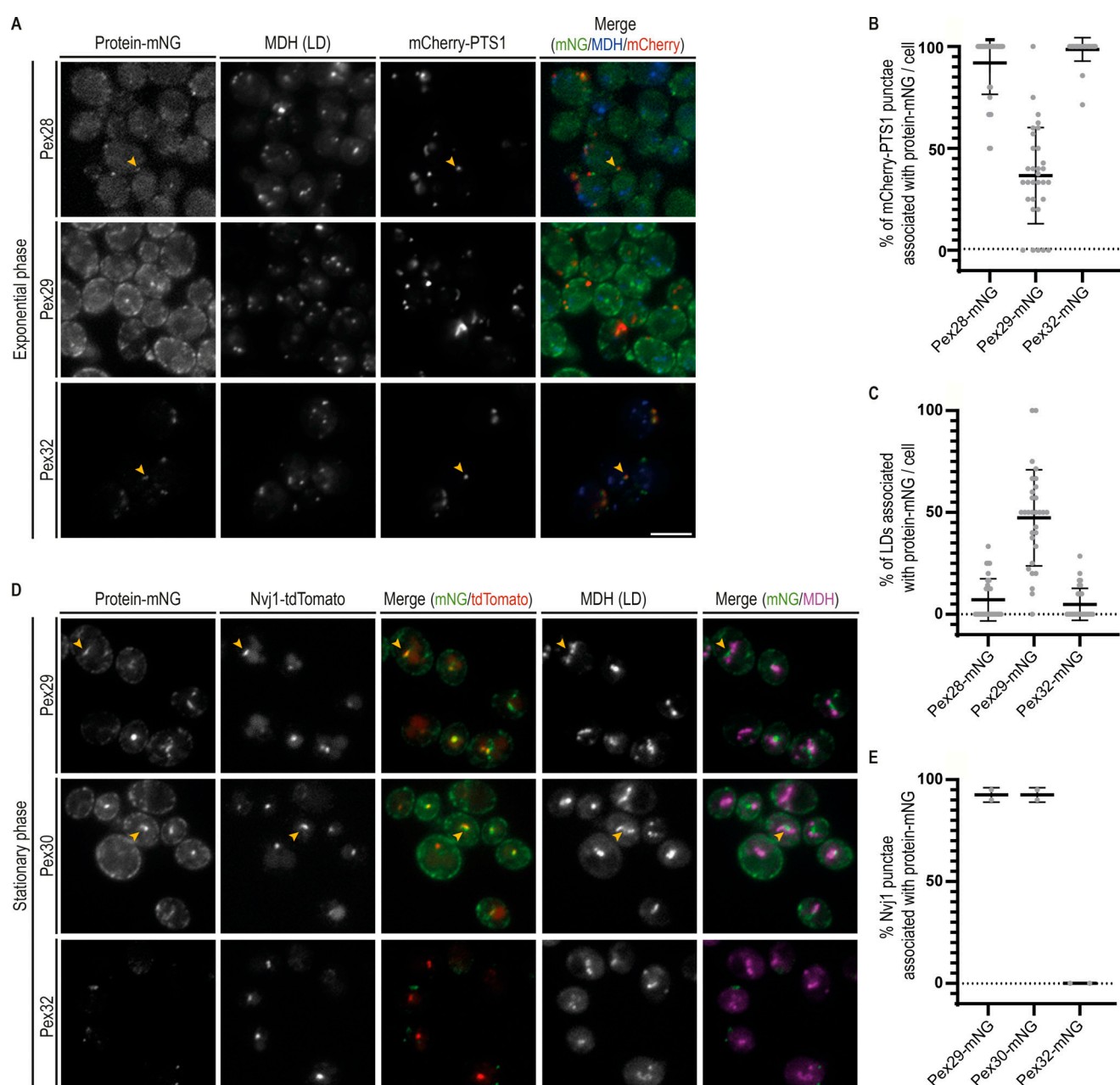

Figure 3. **Distinct ER MCS are targeted by the Pex28/Pex32 and Pex29 complexes. (A)** Localization of Pex28, Pex29, and Pex32 in relation to peroxisomes and LDs. Endogenous Pex28, Pex29, and Pex32 were expressed as fusions to mNG and analyzed in WT cells during exponential growth. Peroxisomes were visualized by mCherry-PTS1, and LDs were stained with the neutral lipid dye MDH. Arrowheads indicate sites of overlap between Pex28- or Pex32-mNG and the peroxisome marker mCherry-PTS1. Bar, 5 μm. **(B)** Colocalization between the peroxisome marker mCherry-PTS1 and Pex28-, Pex29-, and Pex32-mNG was quantified in cells grown as in A. Three independent experiments were analyzed (>30 cells/genotype/experiment were counted). Each dot corresponds to a cell, and bars represent mean and SD. **(C)** Colocalization between MDH-stained LDs and Pex28-, Pex29-, and Pex32-mNG was quantified in cells grown as in A. Three independent experiments were analyzed (>30 cells/genotype/experiment were counted). Each dot corresponds to a cell, and bars represent mean and SD. **(D)** Localization of Pex29, Pex30, and Pex32 in relation to the NVJ. Endogenous Pex29, Pex30, and Pex32 were expressed as fusions to mNG and analyzed in WT cells during early stationary phase. The NVJ was labeled by endogenous Nvj1 expressed as a fusion to tdTomato (Nvj1-tdTomato), and LDs were stained with the neutral lipid dye MDH. Bar, 5 μm. **(E)** Colocalization between the NVJ marker Nvj1-tdTomato and Pex29-, Pex30-, and Pex32-mNG was quantified in cells grown as in D. Three independent experiments were analyzed (>30 cells/genotype/experiment were counted). Each dot corresponds to a cell, and bars represent mean and SD.

RHD, implicated in membrane tubulation (Joshi et al., 2016), is also critical for Pex30 to bind and stabilize its partners.

To further characterize the role of the RHD in binding and stabilizing Pex30 partners, we generated several chimeric proteins (Fig. 5 A). In the first chimera, the RHD of Pex30 was replaced by the one of Pex31 (Pex30$^{Pex31RHD}$), whose stability is independent of Pex30 (Fig. 2 B). The second chimera contained the RHD of Rtn1(Pex30$^{Rtn1RHD}$), which has well-known

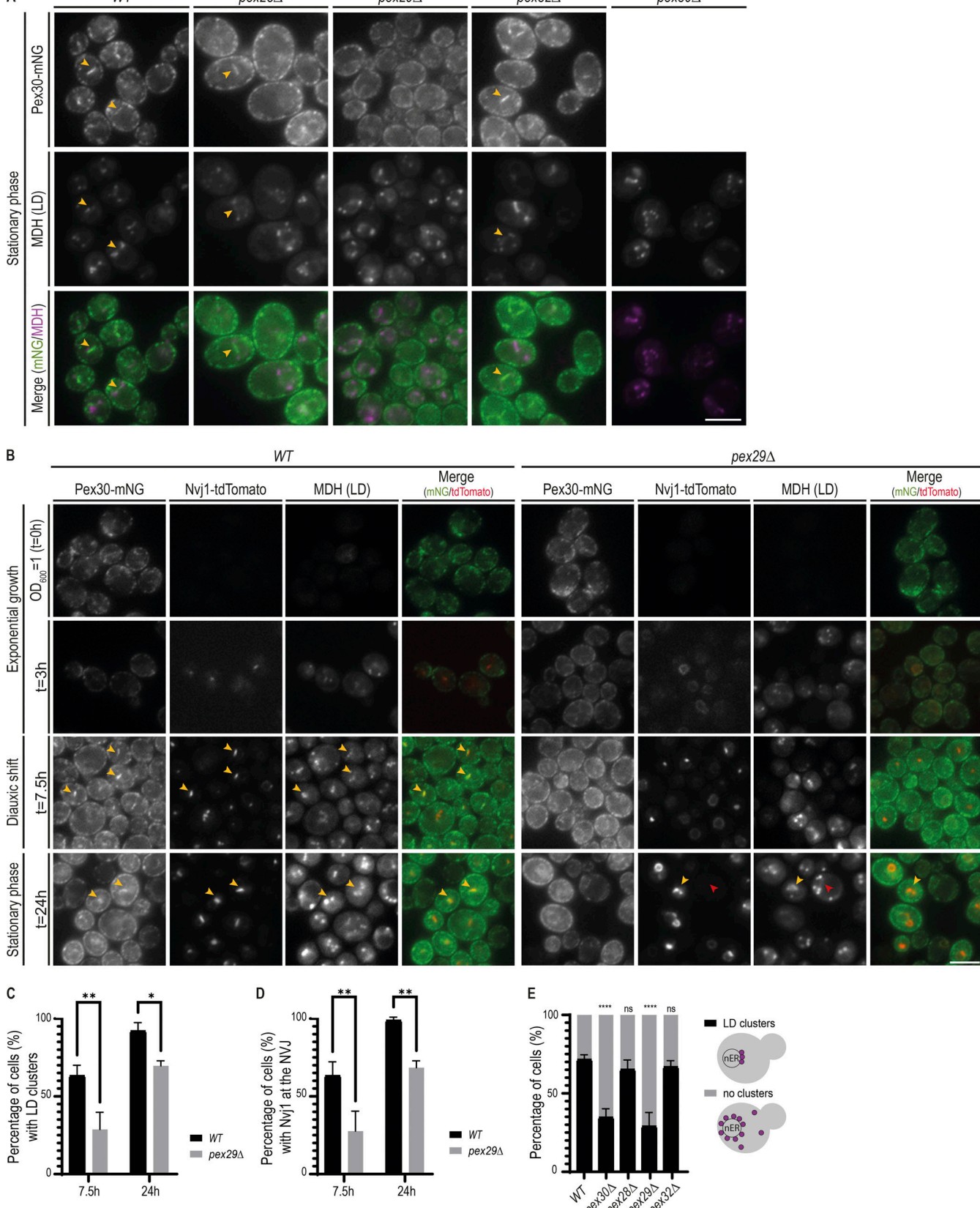

Figure 4. **Pex30/Pex29 complex contributes to spatial organization of LDs at the NVJ. (A)** Localization of Pex30 and LDs in cells with the indicated genotype during early stationary phase. Pex30 was endogenously tagged with mNG (Pex30-mNG) and LDs were stained with the neutral lipid dye MDH. Bar, 5 µm. **(B)** Time-course analysis of Pex30-mNG distribution in WT and *pex29Δ* cells. Exponentially growing cells were monitored through diauxic shift into early

stationary phase as indicated. The formation of NVJ was monitored by endogenous Nvj1-tdTomato, and LDs were stained with the neutral lipid dye MDH. Yellow and red arrowheads indicate cells with and without NVJ-clustered LDs, respectively. **(C)** Quantification of cells with the indicated genotype displaying LDs clustered at the NVJ during diauxic shift (7.5 h) and stationary phase (24 h). Three independent experiments were analyzed (>100 cells/genotype/experiment were counted). Bars represent SD. Ordinary one-way ANOVA and Dunnett's multiple comparisons were done to compare the percentage of cells with the WT condition (**, P < 0.01; *, P < 0.05). **(D)** Quantification of cells with the indicated genotype displaying Nvj1 at the NVJ during diauxic shift (7.5 h) and stationary phase (24 h). Three independent experiments were analyzed (>100 cells/genotype/experiment were counted). Bars represent SD. Ordinary one-way ANOVA and Dunnett's multiple comparisons were done to compare the percentage of cells with the WT condition (**, P < 0.01). **(E)** Quantification of cells with the indicated genotype displaying clustered LDs during early stationary phase. LDs were classified as clustered or nonclustered depending on their distribution, as depicted in the cartoon (on the right). Three independent experiments were analyzed (>100 cells/genotype/experiment were counted). Bars represent SD. Ordinary one-way ANOVA and Dunnett's multiple comparisons were done to compare the percentage of cells with LD clusters with the WT condition (****, P < 0.0001; ns, P > 0.5).

functions in shaping ER tubules (Voeltz et al., 2006). Pex28, Pex29, and Pex32 were strongly destabilized in cells expressing Pex30[Pex31RHD] or Pex30[Rtn1RHD], even if high levels of these chimeric proteins were present in cells (Fig. 5 C). In contrast, the levels of Pex28, Pex29, and Pex32 were restored in pex30Δ cells expressing Pex31[Pex30RHD] chimera, a Pex31 derivative containing the RHD of Pex30 (Fig. 5, A–C). Together, these data indicate that Pex30 RHD is necessary and sufficient to bind and stabilize Pex28, Pex29, and Pex32.

We then asked whether expressing Pex31[Pex30RHD], and thereby restoring the levels of Pex28/Pex32, would rescue the peroxisome defects of pex30Δ mutant. Besides the increase in peroxisome number (Fig. 5 D), we found that pex30Δ cells display a cytosolic pool of mCherry-PTS1, suggesting a defect in protein import into peroxisomes in this mutant (Fig. 5 E). As in pex30Δ cells, cells expressing Pex31[Pex30RHD] showed increased peroxisome number and a cytosolic pool of mCherry-PTS1 (Fig. 5, D and E). Since these cells display normal Pex28 and Pex32 levels (Fig. 5 C), this observation indicates that Pex30 domains besides the RHD are required for its function at ER–peroxisome MCSs.

We also tested whether Pex31[Pex30RHD] would be able to target and function at the NVJ. Despite its correct localization to the ER and the restoration of Pex29 levels, Pex31[Pex30RHD] failed to concentrate at the NVJ during early stationary phase, resulting in a delay in Nvj1 correct localization and LD organization (Fig. 5, F and G). Thus, while essential for the binding to its partners, Pex30 RHD is insufficient for the functions of Pex30 at MCSs.

We next focused on Pex30 DysF domain and investigated whether it is required for Pex30 functions. While a Pex30 mutant lacking the complete DysF domain (Pex30[DysFΔ]) was expressed at normal levels and was able to stabilize its partners Pex28, Pex29, and Pex32 (Fig. S5 B), it displayed a peroxisome defect comparable to that of pex30Δ cells, in terms of both peroxisome number and mCherry-PTS1 import defect (Fig. 5, D and E). Moreover, when expressed from the endogenous locus as an mNG fusion protein, Pex30[DysFΔ] failed to localize to the NVJ during stationary phase (Fig. 5 F). As a consequence, Pex30[DysFΔ]-expressing cells showed defects in NVJ organization, including abnormal Nvj1 localization and LD clustering (Fig. 5 G). Finally, the DysF domain was also essential for Pex30-dependent LD biogenesis in seipin mutants, a function that is independent of its binding partners (Fig. S5, D and E). Thus, the DysF domain is essential for Pex30 function at multiple MCSs.

## Discussion

Here we investigated how the membrane protein Pex30 targets and functions at multiple ER subdomains. We show that Pex30 functions independently of its partners at LD assembly sites. In contrast, the Pex30 functions at ER–peroxisome and NVJ MCSs depend on two distinct biochemical complexes with its family members. With Pex28/Pex32, Pex30 targets ER–peroxisome MCSs, while binding to Pex29 targets Pex30 to the NVJ and contributes to the spatial organization of LDs (Fig. 6). Thus, Pex28/Pex32 and Pex29 serve as specific targeting factors enabling Pex30 to function at different MCS.

Proteins of the Pex30 family associate with the ER membrane through an RHD formed by two pairs of closely spaced transmembrane segments (Joshi et al., 2016). RHDs are also present in other proteins, such as reticulons and receptor expression–enhancing proteins, and in all cases they promote high membrane curvature, as observed in ER tubules or budding PPVs (Hu et al., 2008; Voeltz et al., 2006; Joshi et al., 2016; Wang et al., 2021). RHDs function as dimers, and different dimers induce distinct degrees of curvature, as shown recently for the RHDs of Yop1 and REEP5 (Wang et al., 2021). Pex30 RHD is necessary and sufficient to bind to Pex28/Pex32 and Pex29 (Figs. 5 and S5); therefore, it likely forms heterodimers with the RHDs of these proteins. Since Pex30 can also function independently of its family members, its RHD also likely forms homodimers. Future studies should directly evaluate the oligomeric state of Pex30-containing complexes and how their relative abundance is controlled. Also, it will be interesting to test whether the various Pex30-containing RHD dimers induce distinct degrees of membrane curvature and whether curvature helps to specify Pex30 function at different MCSs.

Targeting Pex30 to MCS may also depend on other domains of Pex28/Pex32 and Pex29 through their interaction with peroxisome and vacuolar components, respectively. Consistent with this possibility, it was recently shown in the methylotrophic yeast H. polymorpha that the localization of Pex32 to ER–peroxisome MCSs required Pex11, a peroxisomal membrane protein (Wu et al., 2020). These two mechanisms are not mutually exclusive, and both membrane curvature and interactions with proteins/lipids of other organelles may be involved in targeting Pex30 to MCS.

Pex30 is also characterized by the presence of a DysF domain. Besides Pex30 family members, the DysF domain is also found in yeast Spo73 (Okumura et al., 2015; Parodi et al., 2015) and in human Myoferlin, TECPR1, and Dysferlin, where it was first

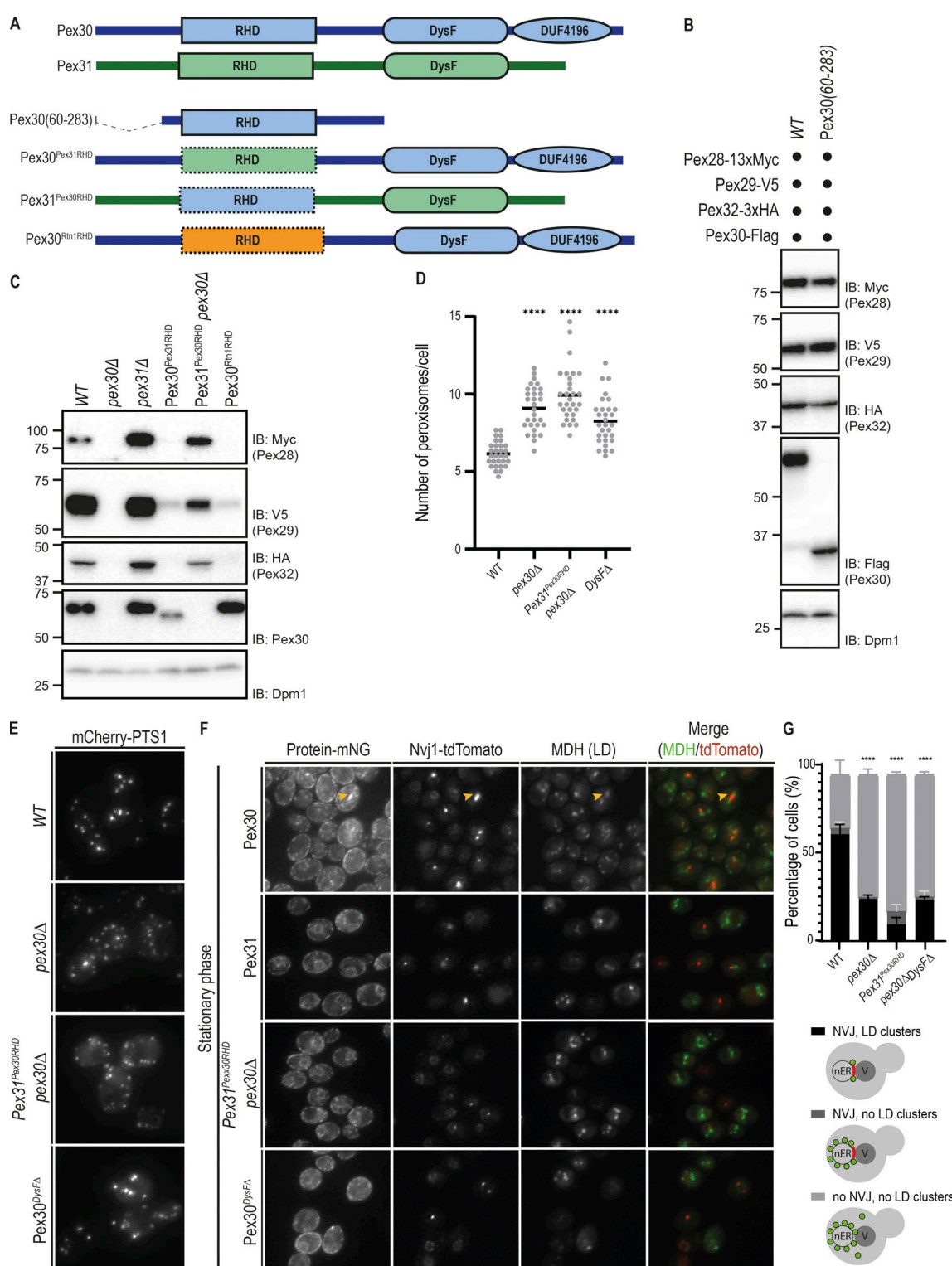

Figure 5. **Pex30 RHD and DysF domains play essential but distinct functions in organelle homeostasis. (A)** Schematic representation of Pex30 (blue), Pex31 (green), and their derivatives. The RHD of Rtn1 is shown in orange. **(B)** Steady-state levels of endogenously tagged Pex28, Pex29, and Pex32 in cells with endogenously expressed Pex30 or Pex30(60–283), a truncated Pex30 derivative containing only the RHD and adjacent regions, as shown in Fig. 5 A. Pex30 or Pex30(60–283) were C-terminally tagged with a FLAG epitope tag and detected with an anti-Flag antibody. Whole-cell extracts were analyzed as in Fig. 1 D. IB, immunoblot. **(C)** Steady-state levels of endogenously tagged Pex28, Pex29, and Pex32 in cells with the indicated genotype. Pex30 and derivatives were detected with an anti-Pex30 antibody. Whole-cell extracts were analyzed as in Fig. 1 D. **(D)** Quantification of the number of peroxisomes per cell, in cells grown as in E. Three independent experiments were analyzed (>30 cells/genotype/experiment were counted). Each dot corresponds to a cell, and bars represent mean. Ordinary one-way ANOVA and Dunnett's multiple comparisons were done to compare the number of peroxisomes with the WT condition (****, P < 0.0001). **(E)** Distribution of peroxisomes in cells with the indicated genotype during exponential growth. Peroxisomes were labeled by the mCherry-PTS1

marker. Please note the increase of cytosolic fluorescence in the mutant cells, corresponding to nonimported mCherry-PTS1. Images correspond to maximum-intensity Z-projections. Bar, 5 µm. **(F)** Localization of Pex30, Pex31, and chimeric constructs in relation to the NVJ. Endogenous Pex30, Pex31, and chimeric constructs were expressed as fusions to mNG and analyzed during early stationary phase. The NVJ was labeled by endogenous Nvj1 expressed as a fusion to tdTomato (Nvj1-tdTomato), and LDs were stained with the neutral lipid dye MDH. An example of NVJ-clustered LDs is indicated (yellow arrowhead). Bar, 5 µm. **(G)** Quantification of cells with the indicated genotype displaying defects in NVJ formation and LD clustering during early stationary phase as in F. Cells were classified in three categories, as depicted in the cartoons: correct NVJ with clustered LDs; correct NVJ but LDs are randomly distributed within the cell; and abnormal NVJ and LDs are randomly distributed within the cell. Three independent experiments were analyzed (>100 cells/genotype/experiment were counted). Bars represent SD. Ordinary one-way ANOVA and Dunnett's multiple comparisons were done to compare the percentage of cells containing correct NVJ and LDs clustered with the WT condition (****, P < 0.0001).

identified (Bulankina and Thoms, 2020; Jeynov et al., 2006). In all these proteins, the DysF domain has been implicated in lipid remodeling processes. Importantly, the majority of DysF pathogenic mutations, responsible for dysferlinopathies in humans, occur within the DysF domain (Sula et al., 2014; Izumi et al., 2020). However, the molecular function of the DysF domain in all these proteins is unknown.

We found that in Pex30, mutations in the DysF domain do not affect protein levels or assembly with its partners. However, the DysF domain is essential for Pex30 function in LD biogenesis, as well as at ER–peroxisome and NVJ MCSs. Given the links of DysF to lipid remodeling processes and the observation that *pex30Δ* cells do not display global changes in lipid composition (Wang et al., 2018), we hypothesize that the DysF domain regulates local lipid composition at ER domains, thereby affecting LD biogenesis as well as other MCSs. Future studies will hopefully clarify whether the Pex30 DysF domain binds directly to lipids and determine its binding specificity and whether this activity is modulated by Pex30 binding partners.

Pex30 and Pex31 have been defined as paralogues because of their common evolution and high level of similarity (Fig. S1 A; Fenech et al., 2020). In fact, their ability to promote ER membrane tubulation may be partly redundant (Joshi et al., 2016). However, our data suggest that Pex30 and Pex31 diverged and mainly carry out distinct functions. In contrast to Pex30, we cannot detect interactions between Pex31 and any of its family members, even though the amino acid sequences of Pex30 and Pex31 RHDs share 72% similarity. Likewise, the DysF domains of Pex30 and Pex31 are nonredundant, as shown by our chimeric constructs between the two proteins. Thus, Pex30 and Pex31 have distinct roles in organelle homeostasis.

While the Pex30/Pex28/Pex32 complex always associates with ER–peroxisome MCSs, the localization of the Pex30/Pex29 complex changes during diauxic shift, concentrating at the NVJ. Upon nutrient exhaustion, the NVJ is formed, and several proteins are known to accumulate at the MCS. The ER membrane protein Nvj1, a prototypical NVJ component, concentrates at this contact site by binding to Vac8 at the vacuolar membrane (Pan et al., 2000). Interestingly, in *pex29Δ* and *pex30Δ* mutants, Nvj1 concentration is delayed, and in some cells it fails to accumulate at the NVJ and remains dispersed throughout the nuclear ER. Although less dramatic, this localization is reminiscent of the one observed in *vac8Δ* cells. Future studies should investigate the defects in Nvj1 localization in *pex29Δ* and *pex30Δ* mutants and whether those defects depend on abnormal Vac8 levels or localization. In WT cells, Pex30/Pex29 also accumulate at the NVJ. How Pex29 defines the localization of the complex upon metabolic reprograming is unknown. Some studies have

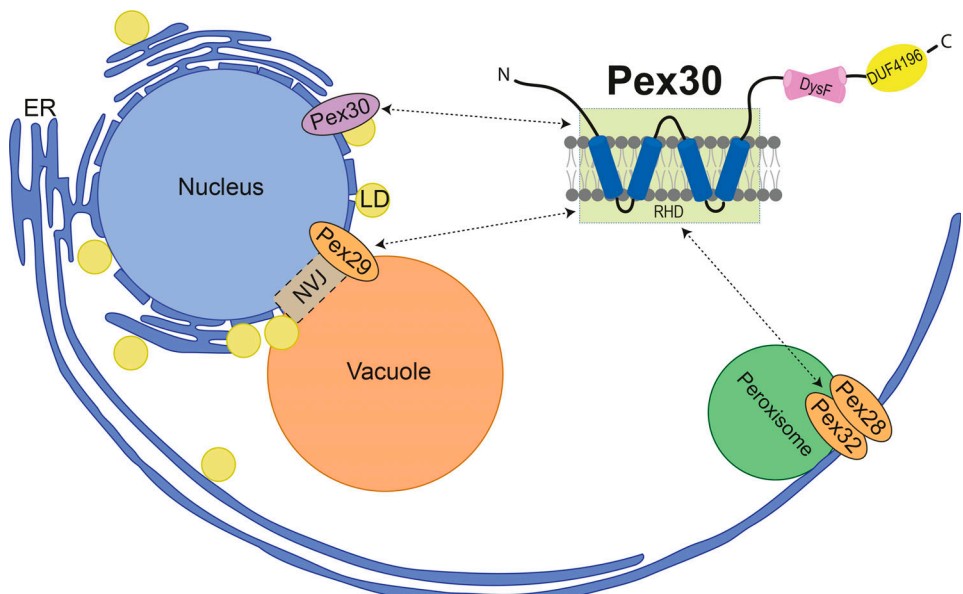

**Figure 6.** **Schematic representation of Pex30 recruitment to distinct MCS by its family member adaptors.** The targeting of Pex30 to diverse MCS depends on the binding of its RHD to its various partners. See text for details.

associated posttranslation modifications with formation of MCSs. For instance, phosphorylation of nonconventional FFAT motifs is required for recognition by the MSP domain of VAP-A/B and MOSPD2 and the formation of MCSs (Di Mattia et al., 2020). Both Pex30 and Pex29 are phosphoproteins (Holt et al., 2009; Albuquerque et al., 2008; Helbig et al., 2010; Martínez-Montañés et al., 2020), but how these modifications contribute to the change in localization has not been explored.

Our data indicate that, among the various family members, Pex30 has a central role, while Pex28, Pex29, and Pex32 act as regulatory subunits to specify Pex30 function at distinct ER domains/MCSs. A similar mode of organization is also observed for Vps13, a soluble lipid transfer protein that functions at multiple MCSs (Li et al., 2020). As in the case of Pex30, Vps13 interacts with different adaptors that facilitate its function at various MCSs (Bean et al., 2018; Kumar et al., 2018; John Peter et al., 2017). Like that of Pex30, Vps13 localization is dynamic and changes in response to cellular metabolism. Thus, this mode of organization appears to be a unifying theme in MCS regulation, perhaps by facilitating a quick adaptation to changing environmental and metabolic cues.

## Materials and methods
### Reagents and antibodies
Pex30 antibody (Joshi et al., 2016) was a kind gift from William Prinz (National Institute of Diabetes and Digestive and Kidney Diseases, National Institutes of Health, Bethesda, MD). Dpm1 antibody (mouse monoclonal 5C5A7; dilution 1:10,000) was purchased from Invitrogen. HA antibody (rat monoclonal 3F10; dilution 1:2,000) was purchased from Roche. V5 antibody (rabbit monoclonal D3H8Q; dilution 1:5,000) was purchased from Cell Signaling. MYC antibody (mouse monoclonal 9E10; dilution 1:1,000) was purchased from Roche. LD dye BODIPY493/503 was purchased from Invitrogen and used at a final concentration of 1 μg/ml. MDH was purchased from Abgent and used at 0.1 mM.

### Yeast strains and plasmids
Yeast strains used in this study are isogenic to either BY4741 (*MATa ura3Δ0 his3Δ1 leu2Δ0 met15Δ0*) or BY4742 (*MATα ura3Δ0 his3Δ1 leu2Δ0 lys2Δ0*) and are listed in Table S1. Tagging of proteins, replacement of promoters, and individual gene deletions were performed by standard PCR-based homologous recombination (Longtine et al., 1998; Janke et al., 2004). Point mutations, protein tagging, replacement of regions within the ORF for chimeric constructs, partial gene deletions, and full gene deletions were performed by CRISPR-based gene editing (adapted from Laughery et al., 2015). Briefly, a single guide RNA sequence targeting the desired region of the gene of interest was designed using the online software Benchling (Biology Software, 2021) and http://wyrickbioinfo2.smb.wsu.edu. The single guide RNA was cloned into the pML107 vector (Laughery et al., 2015), containing a Cas9 endonuclease from *Streptococcus pyogenes*. This plasmid, along with a PCR-amplified template containing the desired modification, was transformed using standard yeast transformation protocol. Strains with multiple deletions/tags

were obtained by crossing haploid cells of opposite mating types, followed by sporulation and tetrad dissection using standard protocols (Fink and Guthrie, 1991). The yeast strains and plasmids used in this study are listed in Table S1 and Table S2, respectively. The oligonucleotides used as template to transform yeast cells with CRISPR-based gene editing are listed in Table S3.

### Growth conditions
Cells were grown at 30°C in YPD liquid medium (1% yeast extract, 2% peptone, and 2% glucose) or synthetic medium (0.67% bacto-yeast nitrogen base without amino acids and 2% glucose), unless indicated otherwise. For microscopy, protein analysis, and immunoprecipitation experiments, exponentially growing cells were analyzed at $OD_{600}$ of 1 or cells were grown to early stationary phase in YPD ($OD_{600}$ of 4–5).

### Protein domain prediction
Sequences were obtained from *Saccharomyces cerevisiae* Genome Database. Domain analysis of Pex30 and its family members was performed using HHpred (Hildebrand et al., 2009).

### Fluorescence microscopy
Fluorescence microscopy was performed at room temperature using the Zeiss Cell Observer HS equipped with a complementary metal–oxide semiconductor (CMOS) camera (HamamatsuORCA-Flash4.0), controlled by 3i Slidebook 6.0 software. A Plan-Apochromat 100× 1.4 objective was used. BODIPY and MDH signals were detected using the GFP and DAPI filters, respectively. mNeonGreen and tdTomato/mCherry signals were detected using the GFP and mCherry filters. LDs were stained with the neutral lipids dyes BODIPY493/503 and MDH. Cells were incubated with a neutral lipid dye for 10 min (1 μg/ml BODIPY and 0.1 mM MDH) at room temperature, spun down, and resuspended in synthetic medium.

### Growth assays
Exponentially growing cells ($OD_{600} \sim 1$) in YPD were diluted to $OD_{600}$ of 0.1. 10-fold serial dilutions were performed in YPD, and 2.5 μl of the dilutions were spotted on YPD agar plates and incubated at 25°C, 30°C, and 37°C for 2–3 d. Photos were taken every 24 h.

### Immunoprecipitation
Immunoprecipitation of endogenously tagged Pex28-13xMyc, Pex29-V5, Pex28-3xHA, Pex29-3xHA, Pex31-3xHA, and Pex32-3xHA was performed as follows. Approximately 100 $OD_{600}$ units of yeast culture grown in YPD were harvested by centrifugation at 3,900 *g*, washed, and resuspended in 700 μl of lysis buffer (LB = 50 mM Tris/HCl, pH 7.4, 200 mM NaCl, 1 mM EDTA, 1 mM phenylmethylsulfonyl fluoride, and complete protease inhibitor [Roche]). Cells were lysed with glass beads, and lysates were cleared by low-speed centrifugation at 4°C. Membranes were pelleted at 45,000 *g* for 25 minutes at 4°C in an Optima Max Tabletop Ultracentrifuge in a TLA 100.3 rotor (Beckman Coulter). The crude membrane fraction was resuspended in 600 μl LB. Then, 700 μl of LB supplemented with decyl maltose neopentyl glycol (DMNG) were added to obtain 1% final concentration

and, membranes were solubilized for 2 h on a rotating wheel at 4°C. Solubilized membranes were cleared for 15 min at 4°C at full speed in a tabletop centrifuge. The tagged proteins were affinity isolated by incubation for 2 h at 4°C with HA magnetic beads (Thermo Scientific), V5 magnetic beads (MBL), and Myc magnetic trap (Chromotek). Beads were washed 3 times with 0.02% DMNG in LB, eluted with Laemmli buffer, and analyzed by SDS-PAGE (PAGE) and immunoblotting. In all experiments, the input lane corresponds to 10% of the total extract used for immunoprecipitation.

### Protein analysis

For Western blotting, whole-cell extracts of exponentially growing cells or cells grown to diauxic shift were prepared from 2 OD units of cells. Pelleted cells were resuspended in 300 μl of 0.15 M NaOH, and incubated on ice for 10 min. After centrifugation at maximum speed for 2 min at 4°C, the pellet was resuspended in sample buffer and heated at 65°C for 10 min. Proteins were separated by SDS-PAGE in Criterion TGX precast gels (Bio-Rad), transferred to PVDF membrane, and analyzed with the indicated antibodies. Antibody signal was acquired by enhanced chemiluminescence (Thermo Fisher Scientific Pierce ECL), and protein levels were quantified using ImageJ.

### Organelle contact site quantification: protein in ER MCS with another organelle

Puncta of Pex30-like proteins were considered to localize to a specific ER–organelle contact site by using all Z planes of Z-stack images. An overlap of the line intensity graphs of a protein of interest and the organelle marker was scored as MCS localization. For each experiment, three sets of a minimum of 30 cells per genotype were analyzed.

### Statistical analysis

Data for all experiments were generated from at least three independent experiments, and no samples were excluded. For microscopy analysis, cells were randomly selected for analysis, and a representative image is shown. For quantification of microscopy data, 300 cells were scored per condition and/or genotype from multiple microscopy fields (unless indicated otherwise in the figure legend). Distributions are presented as mean ± SD. Distributions were tested for normality (Shapiro–Wilk test) and homogeneity of variances (Brown–Forsythe test). Statistical comparisons were made using the appropriate test for the type of data as mentioned in the figure legends (GraphPad Prism 7.0; ****, $P < 0.0001$; ***, $P < 0.001$; **, $P < 0.01$; *, $P < 0.05$; and ns, $P > 0.5$).

### Online supplemental material

Fig. S1 shows that the levels, localization, and function of Pex30 on LD biogenesis are independent of its family members. Fig. S2 shows that protein levels of Pex30 binding partners are dependent on Pex30 availability Fig. S3 shows the localization of Pex30 adaptors in stationary phase. Fig. S4 shows the contribution of the Pex30/Pex29 complex in the formation of nuclear membrane–vacuole junctions in synthetic medium. Fig. S5 shows the Pex30 domains involved in the interaction with Pex28, Pex29, and Pex32 and LD biogenesis. Table S1 lists the yeast strains used in this study. Table S2 lists the plasmids used in this study. Table S3 lists the oligonucleotides used in this study.

## Acknowledgments

We thank S. Wang for preliminary experiments on Pex30 localization to the NVJ. We thank W. Prinz for the anti-Pex30 antibody and R. Klemm, B. Kornmann, L. Krshnan, M. Renne, and J. Robson-Tull for critical reading of the manuscript.

J.V. Ferreira was supported by an Erasmus+ Scholarship, a Saven European Scholarship, and an Oxford-EP Abraham Research Fund Graduate Scholarship. P. Carvalho was supported by a Biotechnology and Biological Sciences Research Council grant (BB/R018375/1) and an investigator award from the Wellcome Trust (202642/Z/16/Z).

The authors declare no competing financial interests.

Author contributions: J.V. Ferreira performed all the experiments; P. Carvalho and J.V. Ferreira conceived the study, analyzed the data, and wrote the manuscript; P. Carvalho supervised the project.

Submitted: 26 March 2021

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

# Supplemental material

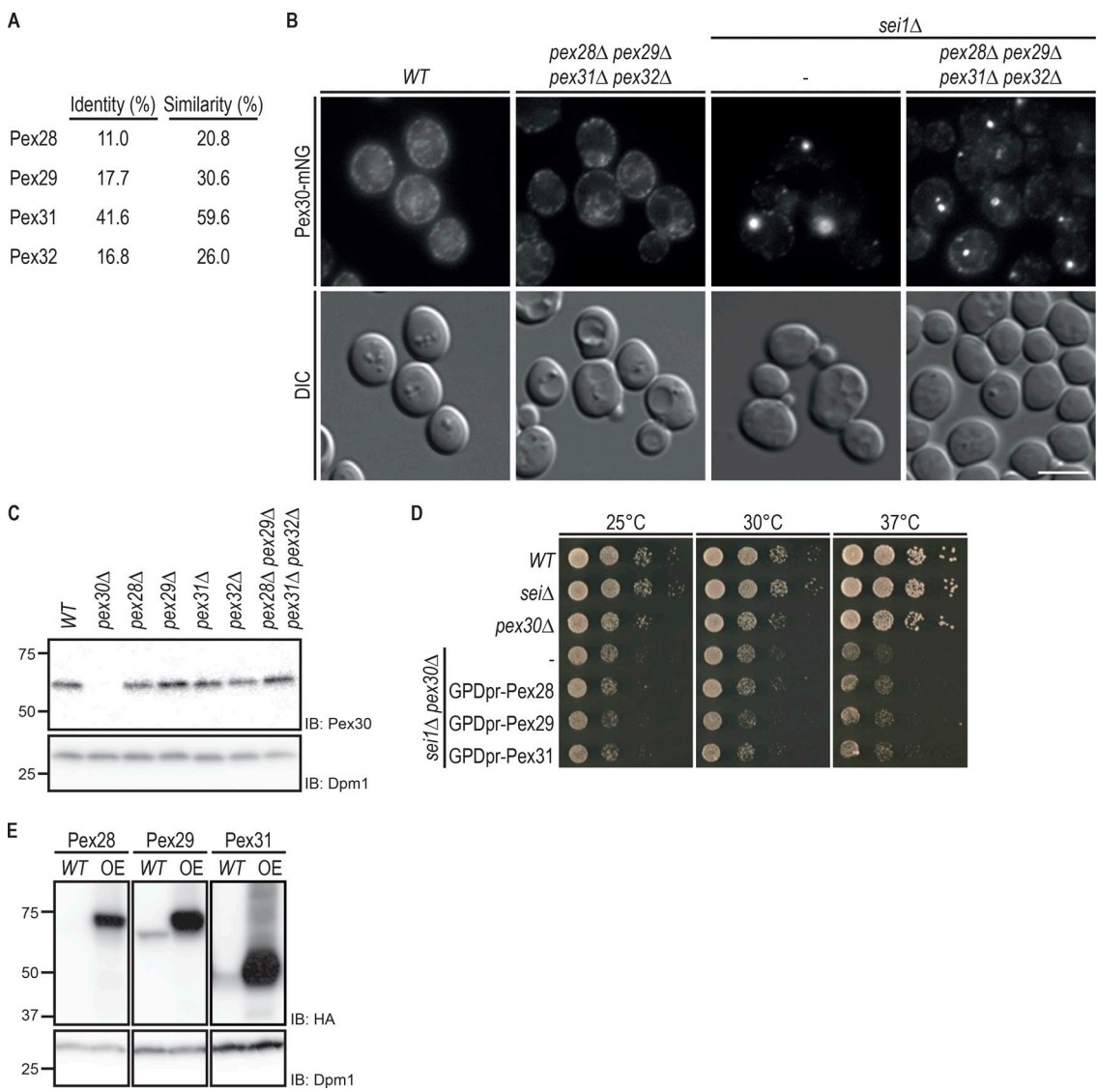

Figure S1.  **Pex30 promotes LD biogenesis independently of its family members. (A)** Percentage of protein sequence identity and similarity between Pex30 and Pex30 family members. EMBOSS Needle – Pairwise Sequence Alignment (PROTEIN) and standard settings were used to analyze the protein sequences. **(B)** Localization of Pex30 in *WT*, *sei1Δ*, and *pex28Δpex29Δpex31Δpex32Δ* cells. Endogenous Pex30 was expressed as a fusion to mNG and analyzed in cells during exponential growth. Bar, 5 µm. DIC, differential interference contrast. **(C)** Steady-state levels of endogenous Pex30 in cells with the indicated genotype during exponential growth. Whole-cell extracts were analyzed as in Fig. 1 D. IB, immunoblot. **(D)** Tenfold serial dilutions of cells with the indicated genotype were spotted on YPD medium and incubated at 25°C, 30°C, or 37°C for 2 d. **(E)** Steady-state levels of endogenous and overexpressed (OE) Pex28, Pex29, and Pex31 in WT cells, during exponential growth phase. Whole-cell extracts were analyzed as in Fig. 1 D.

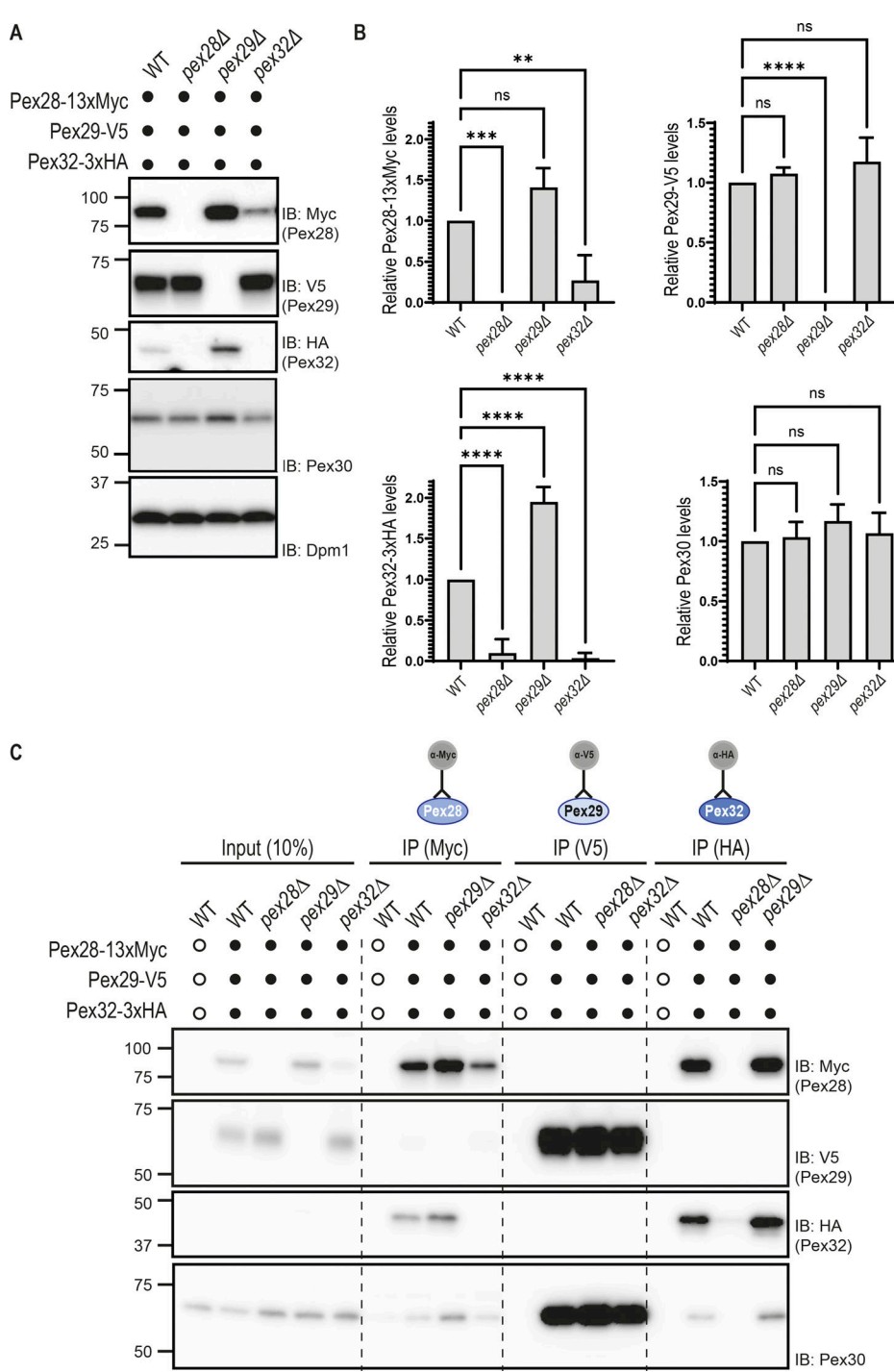

Figure S2.  **Protein levels of Pex30 binding partners are dependent on Pex30 availability. (A)** Steady-state levels of endogenously tagged Pex28, Pex29, Pex30, and Pex32 in cells with the indicated genotype. Whole-cell extracts were analyzed as in Fig. 1 D. Pex28-13MYC, Pex29-V5, Pex32-3HA, Pex30, and Dpm1 (used as loading control) were detected with anti-MYC, anti-V5, anti-HA, anti-Pex30, and anti-Dpm1 antibodies, respectively. IB, immunoblot. **(B)** Quantification of protein levels in cells in the indicated genotype. The levels of Pex30 family proteins determined as in A were compared with WT levels. The average of three independent experiments is shown. Bars represent SD. Ordinary one-way ANOVA and Dunnett's multiple comparisons were used (****, P < 0.0001; ***, P < 0.001; **, P < 0.01; ns, P > 0.5). **(C)** Pex32 interaction with Pex30 is dependent on Pex28. Crude membrane fractions of cells with the indicated genotypes and expressing endogenous Pex28, Pex29, and Pex32 fused respectively, to MYC, V5, and HA epitope tags or untagged proteins as control were detergent solubilized, and extracts were subjected to immunoprecipitation (IP) with anti-MYC, V5, or HA antibodies. Eluted proteins were analyzed by Western blotting.

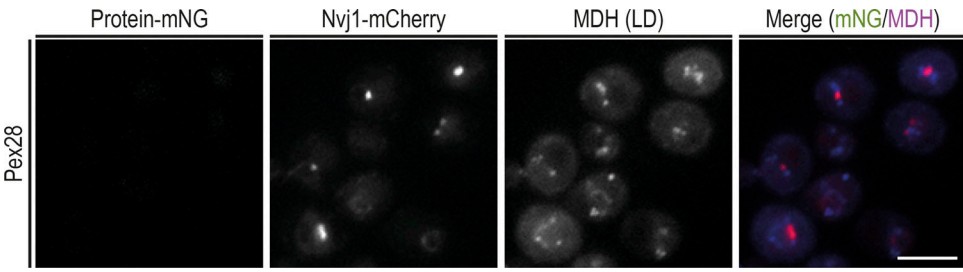

**Figure S3.** **Localization of Pex28 during early stationary phase.** Localization of Pex28 in relation to the NVJ. Endogenous Pex28 was expressed as a fusion to mNG and analyzed in WT cells during early stationary phase. Under these conditions, Pex28 levels are very low, and Pex28-mNG is barely detected. The NVJ was labeled by endogenous Nvj1 expressed as a fusion to tdTomato (Nvj1-tdTomato), and LDs were stained with the neutral lipid dye MDH. Bar, 5 μm.

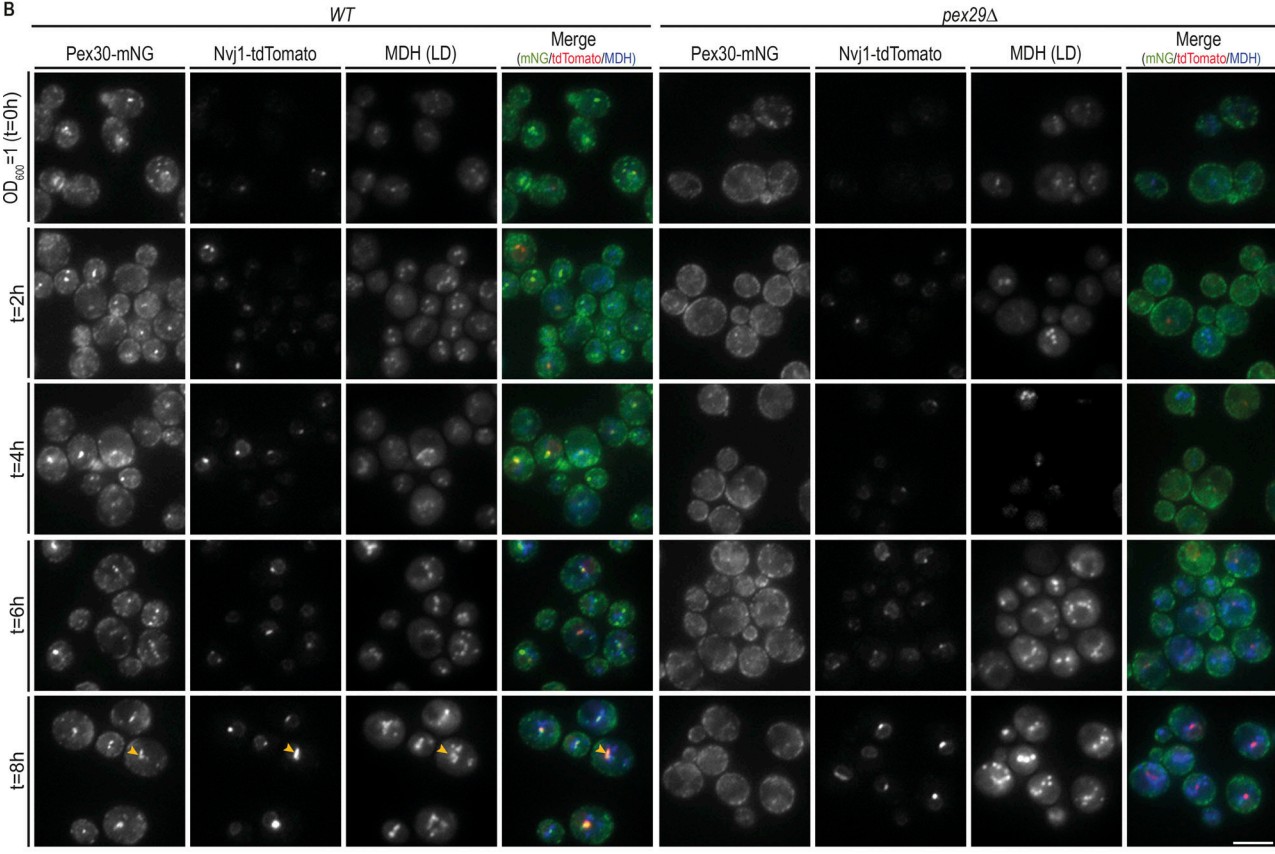

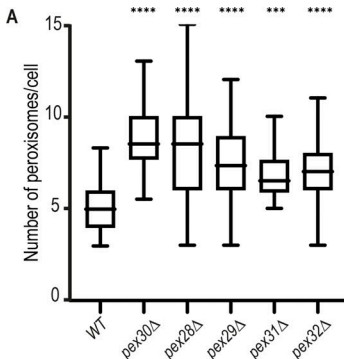

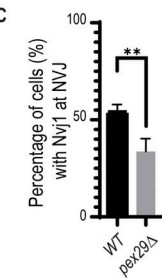

Figure S4. **Pex30/Pex29 role on NVJ formation. (A)** Quantification of the number of peroxisomes per cell, in exponentially growing cells. Three independent experiments were analyzed (>30 cells/genotype/experiment were counted). Box and whiskers represent distribution of the values (minimum, 25th percentile, median, 75th percentile, and maximum). Ordinary one-way ANOVA and Dunnett's multiple comparisons were done to compare the number of peroxisomes with the WT condition (****, P < 0.0001; ***, P < 0.001). **(B)** Time-course analysis of Pex30-mNG distribution in WT and *pex29Δ* cells grown in synthetic medium. Exponentially growing cells were monitored through diauxic shift into early stationary phase as indicated. The formation of NVJ was monitored by endogenous Nvj1-tdTomato, and LDs were stained with the neutral lipid dye MDH. Yellow arrowheads indicate cells with NVJ-clustered LDs. **(C)** Quantification of cells with the indicated genotype displaying Nvj1 localized to the NVJ during diauxic shift (4 h). Three independent experiments were analyzed (>30 cells/genotype/experiment were counted). Bars represent SD. Ordinary one-way ANOVA and Dunnett's multiple comparisons were done to compare the percentage of cells with the WT condition (**, P < 0.01).

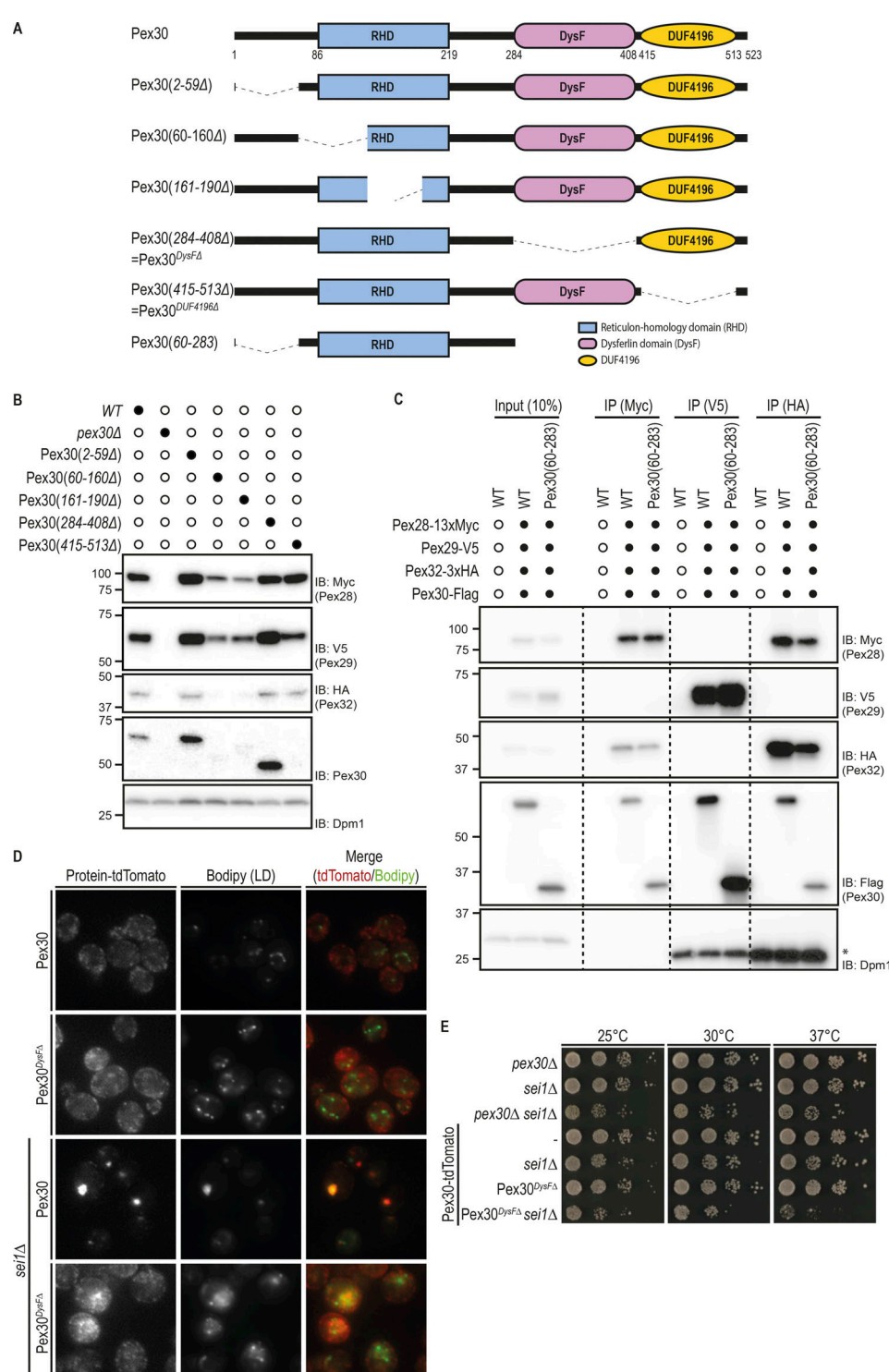

Figure S5. **Pex30 RHD and DysF domains play essential but distinct functions in organelle homeostasis. (A)** Schematic representation of Pex30 and its derivatives with the indicated deletions. **(B)** Steady-state levels of endogenously tagged Pex28, Pex29, and Pex32 in cells with the indicated Pex30 derivatives. Whole-cell extracts were analyzed as in Fig. 1 D. Pex28-13xmyc, Pex29-V5, Pex32-3xHA, Pex30, and Dpm1 (used as loading control), were detected with anti-MYC, anti-V5, anti-HA, anti-Pex30, and anti-Dpm1 antibodies, respectively. Note that Pex30*DUF4196Δ* is not recognized by the anti-Pex30 antibody. This antibody recognizes an epitope at the very C-terminus of Pex30 (Joshi et al., 2016) and that is absent in the Pex30*DUF4196Δ* mutant. IB, immunoblot. **(C)** Pex30 interaction with Pex28/Pex32 and Pex29 only requires Pex30 RHD. Crude membrane fractions of cells with the indicated endogenous Pex30 variants fused to FLAG tag and expressing endogenous Pex28, Pex29, and Pex32 fused respectively, to MYC, V5, and HA epitope tags or untagged proteins as control were detergent solubilized, and extracts were subjected to immunoprecipitation (IP) with anti-MYC, V5, or HA antibodies. Eluted proteins were analyzed by Western blotting. **(D)** Localization of Pex30 and Pex30*DysFΔ*, a Pex30 internal mutant, in WT and *sei1Δ* cells. Both Pex30 and Pex30*DysFΔ* were expressed from the endogenous Pex30 locus as a fusion to tdTomato fluorescent protein. LDs were stained with the neutral lipid dye BODIPY 493/503. Bar, 5 µm. **(E)** Tenfold serial dilutions of cells with the indicated genotype were spotted on YPD medium and incubated at 25°C, 30°C, or 37°C for 2 d.

**Provided online are three tables. Table S1 lists the yeast strains used in this study. Table S2 lists the plasmids used in this study. Table S3 lists the oligonucleotides used in this study.**

