## [Peer Review File · The Journal of Cell Biology]

Pex30-like proteins function as adaptors at distinct ER membrane contact-sites

Joana Ferreira and Pedro Carvalho

Corresponding Author(s): Pedro Carvalho, University of Oxford

Review Timeline:

Submission Date:	2021-03-26
Editorial Decision:	2021-05-06
Revision Received:	2021-06-19
Editorial Decision:	2021-07-13
Revision Received:	2021-07-19

Monitoring Editor: Jodi Nunnari

Scientific Editor: Andrea Marat

Transaction Report:

DOI: <https://doi.org/10.1083/jcb.202103176>

May 6, 2021

Re: JCB manuscript #202103176

Dr. Pedro Carvalho
University of Oxford
Sir William Dunn School of Pathology
South Parks road
South Parks Road
Oxford, UK OX1 3RE
United Kingdom

Dear Dr. Carvalho,

Thank you for submitting your manuscript entitled "Pex30-like proteins function as Pex30 adaptors at distinct ER membrane contact-sites". The manuscript was assessed by expert reviewers, whose comments are appended to this letter. We're very sorry for the delay in communicating this decision to you. We invite you to submit a revision if you can address the reviewers' key concerns, as outlined here.

You will see that the reviewers are all positive about the study -- they (and we agree) found your analyses of Pex30 complexes and their specificity to MCSs very interesting. The reviewers have a number of points that we editorially find on-point, constructive, and valid. Many require clarifications of the phenotypes and interpretation, including clarity on the LD analyses in the mutant cells that open the paper (Rev#2 #1-2, see also Rev#3 point #1) and data interpretation suggestions (Rev#2 #3-4-5). Rev#3, like Rev#2, was skeptical of the deletion studies determining the regions important for Pex30 to bind to its partners (#2) and wanted more quantitated phenotypic studies of the impact of the truncated proteins on the architecture of contacts (#3). We recommend that you address all the reviewers' remarks to the best of your ability and would be happy to discuss the reviews further if you anticipate any issues tackling them or have any questions.

GENERAL GUIDELINES:

Text limits: Character count for an Article is < 40,000, not including spaces. Count includes title page, abstract, introduction, results, discussion, acknowledgments, and figure legends. Count does not include materials and methods, references, tables, or supplemental legends.

Figures: Articles may have up to 10 main text figures. Figures must be prepared according to the policies outlined in our Instructions to Authors, under Data Presentation, <https://jcb.rupress.org/site/misc/ifora.xhtml>. All figures in accepted manuscripts will be screened prior to publication.

***IMPORTANT: It is JCB policy that if requested, original data images must be made available.

Failure to provide original images upon request will result in unavoidable delays in publication. Please ensure that you have access to all original microscopy and blot data images before submitting your revision.***

Supplemental information: There are strict limits on the allowable amount of supplemental data. Articles may have up to 5 supplemental figures. Up to 10 supplemental videos or flash animations are allowed. A summary of all supplemental material should appear at the end of the Materials and methods section.

As you may know, the typical timeframe for revisions is three to four months. However, we at JCB realize that the implementation of social distancing and shelter in place measures that limit spread of COVID-19 also pose challenges to scientific researchers. Lab closures especially are preventing scientists from conducting experiments to further their research. Therefore, JCB has waived the revision time limit. We recommend that you reach out to the editors once your lab has reopened to decide on an appropriate time frame for resubmission. Please note that papers are generally considered through only one revision cycle, so any revised manuscript will likely be either accepted or rejected.

Thank you for this interesting contribution to the Journal of Cell Biology. You can contact us at the journal office with any questions, cellbio@rockefeller.edu or call (212) 327-8588.

Sincerely,

Jodi Nunnari, Ph.D.
Editor-in-Chief, Journal of Cell Biology

Melina Casadio, Ph.D.
Senior Scientific Editor, Journal of Cell Biology

Reviewer #1 (Comments to the Authors (Required)):

Peroxisins, or Pex proteins, are proteins that are required for peroxisome assembly. But many Pex proteins have roles outside of peroxisome assembly and which are accomplished to a large degree through different membrane contact sites (MCSs) formed with different membrane-bounded organelles. However, little is known how some Pex proteins can be delivered to different sites in the cell to perform functions other than the one they were initially ascribed in peroxisome biogenesis. One group of Pex proteins with different locations in the cell and with different functions in the cell depending on where they localize in the cell are the Pex30 family of proteins, made up of Pex30 itself together with Pex28, Pex29, Pex31 and Pex32. In this manuscript, Ferreira and Carvalho report how different complexes of Pex30 family members target and function at different MCSs in cells of the yeast *Saccharomyces cerevisiae*. The authors combine fluorescence microscopy and pull-downs of protein complexes to show that Pex30 joins Pex28 and Pex32 to target to the ER-

peroxisome MCS, Pex30 joins Pex29 to organize the nuclear-vacuolar junction (NVJ), while Pex30 on its own promotes lipid droplet biogenesis. The Pex30 family members, like reticulon proteins, shape membranes through their reticulon homology domains (RHDs), and the authors show that the RHDs of Pex30 family members mediate the assembly of the various complexes through an analysis of Pex30 family member mutants deleted for the RHD domain.

This manuscript makes an important contribution to our understanding how Pex proteins, particularly Pex30 protein family members, perform a variety of roles in the cell outside peroxisome biogenesis by complexing together with different family members, which in turn targets them to different MCSs in the yeast cell and allows them to perform a variety of functions at these sites. The experiments are comprehensive, and the results convincingly support the conclusions of the paper. The authors should address the following issues:

- 1) p. 8. The authors say that overexpression of Pex28, Pex29 or Pex31 failed to revert LD biogenesis in *sei1Δpex30Δ* cells (Figure 1F). However, it looks that in fact there is partial LD biogenesis recovery in these cells when Pex29 or Pex31 is overexpressed. The authors should consider rephrasing.
- 2) p. 10, 2nd paragraph. The authors state that cells lacking Pex29 have higher levels of both Pex28 and Pex32. However, from the images in Figure S2B, the increase in levels of Pex28 in *pex29Δ* cells is not significant (ns). The authors should rephrase their statement on p. 10.
- 3) p. 11, 5-4th lines from bottom. The authors state that Figure 3D shows that "...Pex29 and Nvj1 showed a complete overlap...". This is an over statement. There is extensive, but still only partial, overlap. The authors should rephrase and tone down their statement.
- 4) p. 12. There is no need to call it Figure S3A. Figure S3 is fine. Also, remove the 'A' in the figure itself.
- 5) p. 14, 3rd line from bottom. Figure 5D not Figure 5C.
- 6) p. 14, last line. Figure 5C not Figure 5D.
- 7) Figure 4E. Describe what the cartoon is supposed to tell the reader.
- 8) p. 41. Figure S4A legend. What does *** signify?
- 9) p. 41. Figure S4B legend. There are no red arrowheads. Correct.
- 10) 'data' is plural, 'datum' is singular. 'media' is plural, 'medium' is singular. Correct throughout the manuscript.
- 11) Throughout the manuscript, the number of a verb does not match the number of its subject. For example, on p. 5, 4th line from bottom, the authors write "...the function of Pex30 depend on..." when of course it should be "...the function of Pex30 depends on..." There are a number of similar errors throughout the manuscript. The authors must correct.

Reviewer #2 (Comments to the Authors (Required)):

In this manuscript Ferreira and Carvalho expand upon a previous observation in the Carvalho lab that Pex30 and seipin collaborate to mark sites on the ER for lipid droplet (LD) and peroxisome assembly and participate in these processes. Pex30 is similar in structure to Pex28, Pex29, Pex31, and Pex32, and the authors ask whether they perform similar functions or form complexes. The find that Pex30 is unique in that the $pex30\Delta sei1\Delta$ phenotype is not reproduced by substituting the others for Pex30. They then do a series of pull-down experiments and show that Pex31 does not participate, while Pex30 forms a complex with Pex29 and another complex with Pex28 and Pex32, with Pex28 necessary for Pex30 to bind to Pex32. The group finds that Pex28 and Pex32 can colocalize with peroxisomal structures, while Pex29 binds to the NVJ. In fact, Pex29 is essential for the NVJ structure. Next, the authors probe domains in Pex30 responsible for interactions with the other family members by making deletion mutants. They find that the absence of Pex30 destabilizes the other members (except Pex31?) and that the reticulon-homology domain (RHD) may be important for stabilizing the other proteins. The dysF domain, in contrast, is required for function at the NVJ and peroxisomal sites.

I liked this paper, as it points to a more general role of the Pex30 family members in several inter-organelle junctions. I found the involvement of Pex29-Pex30 in NVJ structure and function particularly compelling. The biochemistry also clearly shows Pex30 in complexes with the others, with the exception of Pex31 which appears to act as a lone wolf. There is important information herein on relating elements of structure to binding in the complexes vs. nurturing at the organelle junctions. However, I stumbled through the interpretations of several key experiments, and I think these need clarification if not other controls and possible re-evaluation before I would advise acceptance in this journal:

1) Figure 1B: The figure clearly shows a difference in $pex30\Delta$ phenotypes in combination with $sei1\Delta$, compared with the other Pex proteins in the family. The authors imply that the diffuse BODIPY staining in the $sei1\Delta pex30\Delta$ strain indicates the inability of these cells to assemble lipid droplets. They have not shown this; they have only shown a lack of LDs at steady state. This strain has a growth defect, and one interpretation of the result is that the cells have used up all of their neutral lipid in energy metabolism (some compensatory mechanism?), and this has nothing to do with LD assembly. Perhaps peroxisomes are more active in beta-oxidation in this strain? Is triglyceride (TG) lipolysis increased? Is the concentration of TG different from the other strains in the panel? If there is plenty of neutral lipid, shouldn't BODIPY stain the ER if neutral lipids are backed up into that organelle?

2) Also Figure 1B: There are small LDs in the double deletions of $sei1\Delta$ with $pex28\Delta$, $pex29\Delta$, and $pex31\Delta$, compared with $sei1\Delta$ alone. The authors might comment on this. On the surface it seems that Pex28, Pex29, and Pex31 may normally suppress droplet formation, which would be quite interesting.

3) Figure 2A: In the text the authors write, "...Pex28, Pex29, and Pex32 efficiently co-precipitated with Pex30." I agree this is true for Pex29, if you compare Pex30 input with Pex30 pulled-down. For the Pex28 and Pex32, only ~20% of Pex30 is pulled down, and it is impossible to know the stoichiometry of Pex30 to the others. So I'd simply omit "efficiently." By the way, as Pex28 mediates the binding of Pex30 to Pex32 (shown later), but Pex32 pulls down Pex30, the authors can further conclude from Fig. 2A that much (most if not all?) of Pex28 is constitutively bound to Pex30.

4) Figure 3A, 3B: The authors state that "all Pex28 and Pex32 foci were apposed to a peroxisome. . .", but 3B shows the converse, that all peroxisomes were associated with Pex28 (or Pex32) dots. In Fig. 3A, many green dots (in the merge) do not have corresponding red dots.

5) Figure S5B: This figure is hard to interpret, as only WT, 2-59 Δ and 284-408 Δ are expressed, according to the immunoblot. The pull-downs then show that the amino terminus and the DysF domain are not important for association with the other Pex proteins. But I don't think you can conclude anything about the RHD from this experiment. The fact that significant Pex28 and Pex29 are pulled down by invisible RHD deletions show, if anything, that the intact RHD domain is dispensable for the protein associations, not that "RHD. . . is critical for Pex30 to bind and stabilize its partners."

6) Figure 5: Along the same lines, the authors swap out RHD from Pex30 with RHDs from other Pex proteins. They don't work for binding partners, leading to their conclusion that, "Pex30 RHD is necessary and sufficient to bind and stabilize Pex28, Pex29, and Pex32." It may be necessary (but see my comment (5) above), but it is only sufficient in the context of the other domains in the other Pex proteins, as these are present in the chimeras. It seems very possible that one needs BOTH Pex30 RHDs AND other elements found downstream in all the PEX proteins of the family. "Sufficient" is too strong a term.

Reviewer #3 (Comments to the Authors (Required)):

Ferreira and Carvalho investigate the relationship between Pex30 and members of the Pex30 family, which includes Pex28, Pex29, Pex31, and Pex32. They find that Pex30's function in lipid droplet (LD) biogenesis does not depend on the other members of the family. Interestingly, they show that Pex30 assembles with different family members into distinct complexes that target different MCSs. Pex30 functions in a complex with Pex28 and Pex32 that targets ER-peroxisome membrane contact sites (MCSs), and a complex of Pex30 and Pex29 localizes to the nuclear vacuole junction (NVJ). Pex29 is required for the localization of Pex30 to the NVJ. In the absence of Pex29, NVJ organization, including LD clustering at the NVJ, is disrupted. Using a structure-function approach, they go on to show that the RHD and DysF domain are important for Pex30 function at multiple MCSs. The RHD is important for the formation of both Pex30 complexes, while the DysF domain is required for functions of Pex30, perhaps in lipid remodeling, that are important to peroxisome and LD biogenesis and NVJ organization.

The characterization of the two distinct Pex30 complexes at both the biochemical and cell biological levels is well done. The findings are well-presented and would be of interest to the MCS field. Prior to publication a few points need to be addressed.

Major comments:

1. In Fig. 1, the authors present data demonstrating that the function of Pex30 in LD biogenesis does not require the other Pex30 family members. They then go on to overexpress the other family members, which are less abundant in cells, to show that when overexpressed, they cannot substitute for Pex30 in LD biogenesis. However, in Fig. 1F, punctate BODIPY-labeled structures can be seen in cells overexpressing Pex29 and to a lesser extent Pex31. Are these structures LD, and, if so, would these results suggest a partial rescue? In addition, the authors should quantify the LD phenotypes observed for the experiments shown in Fig. 1.

2. The authors use results obtained with mutants that have partial deletions of the RHD to support their conclusion that the RHD is critical for Pex30 to bind and stabilize its binding partners. However, the Pex30 RHD mutants are either unstable or expressed at levels far lower than WT Pex30. There

are no bands visible/extremely faint bands for these proteins in the western blot shown in Fig. S5. Therefore, it is difficult to conclude, based on these data, that the RHD is critical for Pex30 to bind its partners as the RHD Pex30 mutants may be misfolded or completely destabilized. This needs to be addressed when discussing Fig. S5B.

The chimera data do help support their conclusion regarding the RHD. Is expression of the Pex30 RHD domain alone sufficient to bind and stabilize its binding partners or are the chimeras necessary for proper localization, stability, etc?

The DUF mutant is also not visible in the western blot shown in Fig. S5. It is likely due to the fact that the Pex30 antibody was made against AAs 509-519, which are missing in that mutant. The authors need to make this clear in the legend, or else it is very confusing to the reader.

3. In Fig 5E, are the Pex mutants not localizing to the NVJ or is the NVJ not being formed/delayed formation? For Fig. 5E, the legend describes that the cells shown are also labeled with NVJ and LD markers, however only Pex protein localization is shown. The authors need to show the NVJ and LD markers in the figure along with the Pex protein localization. In addition, the authors need to add quantification for the percentage of cells in which NVJs are visible as well as quantification of Pex protein localization to the NVJ. For 5F, a better description of the quantification is needed. Does the quantification only include cells that have an NVJ or are all cells included? Three categories should be quantified: no NVJ, NVJ with no LD cluster, NVJ with LD cluster. A similar analysis should also be done for Fig. 4E.

Minor comments:

1. It would be very useful to better describe the LD phenotype for the *sei1* mutant in the results section for readers less familiar with the field. It would make it easier to interpret the LD images in Fig. 1.

2. For the figures in which stationary phase cells are shown, it would be very helpful to the reader to indicate within the figure itself which panels are showing stationary phase cells - for example, 3D, 4A, and 5E. It is stated in the legend but would also be useful if indicated in the figure.

3. The authors use Fig. 4A to conclude that Pex30 fails to localize to the NVJ in a *pex29* mutant. They provide clear evidence of this in Fig 4B, but Fig. 4A cannot be used to make such a definitive statement as they are not looking at an NVJ marker. The wording regarding the conclusion to Fig. 4A needs to be modified. Also, in Fig. 4D, what is meant by Nvj1 clusters? Is it localization of Nvj1 to an NVJ? -The word cluster here is confusing.

4. The legend for Fig. 4C refers to quantifications in 4B but there are no quantifications in 4B.

5. In the results section that describes Fig. 5 (end of page 14), panels C and D are misreferenced.

Response to the reviewers

Reviewer 1:

1-“p. 8. The authors say that overexpression of Pex28, Pex29 or Pex31 failed to revert LD biogenesis in *sei1Δpex30Δ* cells (Figure 1F). However, it looks that in fact there is partial LD biogenesis recovery in these cells when Pex29 or Pex31 is overexpressed. The authors should consider rephrasing”

We thank the reviewer for this comment and suggestion. We apologize that our initial text did not accurately described the phenotype of *sei1Δpex30Δ* cells, which display unique dispersed BODIPY staining. As described previously (Wang et al., 2018), these structures overlap with the ER and likely correspond to the accumulation of neutral lipids in the ER membrane due inefficient LD budding. We have now changed the text to better describe our results. In particular, that the unique dispersed BODIPY staining of *sei1Δpex30Δ* cells is not altered upon Pex28, Pex29 or Pex31 overexpression.

2-“p. 10, 2nd paragraph. The authors state that cells lacking Pex29 have higher levels of both Pex28 and Pex32. However, from the images in Figure S2B, the increase in levels of Pex28 in *pex29Δ* cells is not significant (ns). The authors should rephrase their statement on p. 10”

The text has been changed to better describe the data.

3-“p. 11, 5-4th lines from bottom. The authors state that Figure 3D shows that “...Pex29 and Nvj1 showed a complete overlap...”. This is an over statement. There is extensive, but still only partial, overlap. The authors should rephrase and tone down their statement.”

We meant to say that the Pex29 brighter structures completely overlap with Nvj1. The text has now been changed.

4-“p. 12. There is no need to call it Figure S3A. Figure S3 is fine. Also, remove the 'A' in the figure itself.”

The text, the figure and figure legend have been changed.

5- "p. 14, 3rd line from bottom. Figure 5D not Figure 5C."

This has been fixed.

6-"p. 14, last line. Figure 5C not Figure 5D."

This has been fixed.

7-"Figure 4E. Describe what the cartoon is supposed to tell the reader. "

This has been fixed.

8-"p. 41. Figure S4A legend. What does * signify?"**

This has been fixed.

9- "p. 41. Figure S4B legend. There are no red arrowheads. Correct."

This has been fixed.

10-"data' is plural, 'datum' is singular. 'media' is plural, 'medium' is singular. Correct throughout the manuscript."

We did our best to fix the grammar throughout the manuscript.

11- "Throughout the manuscript, the number of a verb does not match the number of its subject. For example, on p. 5, 4th line from bottom, the authors write "...the function of Pex30 depend on..." when of course it should be "...the function of Pex30 depends on..." There are a number of similar errors throughout the manuscript. The authors must correct."

We did our best to fix the grammar throughout the manuscript.

Reviewer 2:

1- “Figure 1B: The figure clearly shows a difference in *pex30Δ* phenotypes in combination with *sei1Δ*, compared with the other Pex proteins in the family. The authors imply that the diffuse BODIPY staining in the *sei1Δpex30Δ* strain indicates the inability of these cells to assemble lipid droplets. They have not shown this; they have only shown a lack of LDs at steady state. This strain has a growth defect, and one interpretation of the result is that the cells have used up all of their neutral lipid in energy metabolism (some compensatory mechanism?), and this has nothing to do with LD assembly. Perhaps peroxisomes are more active in beta-oxidation in this strain? Is triglyceride (TG) lipolysis increased? Is the concentration of TG different from the other strains in the panel? If there is plenty of neutral lipid, shouldn't BODIPY stain the ER if neutral lipids are backed up into that organelle? “

We thank the reviewer for these comments and apologize for the incomplete description of the phenotype of *sei1Δpex30Δ* cells in the original version of the manuscript. In a previous study (Wang et al., 2018), we characterized in detail the dispersed BODIPY structures unique to *sei1Δpex30Δ* cells. In these cells, the biogenesis of both LDs and peroxisomes was very inefficient. Using lipidomics we also showed that *sei1Δpex30Δ* cells have fairly normal levels of TAG and SE while the levels of the major phospholipids (PC and PI) are greatly increased. This resulted in cells with massively expanded ER where neutral lipids appeared to be trapped - this is precisely what the dispersed BODIPY structures observed in *sei1Δpex30Δ* cells are (see Figure 1B and 1F). In addition to the dispersed BODIPY structures, we also see a few brighter foci, which likely correspond to a few aberrant LDs that still form in these cells. Both structures (dispersed and brighter foci) were characterized by thin section EM in Wang et al., 2018.

2-“Also Figure 1B: There are small LDs in the double deletions of *sei1Δ* with *pex28Δ*, *pex29Δ*, and *pex31Δ*, compared with *sei1Δ* alone. The authors might comment on this. On the surface it seems that Pex28, Pex29, and Pex31 may normally suppress droplet formation, which would be quite interesting.”

Yes, we agree with the reviewer. In *sei1Δ* cells, LDs are very heterogeneous, with tiny LDs interspersed with a few supersized LDs. This phenotype has been extensively characterized by many groups and the relative abundance of tiny and supersized LDs depends on the growth conditions/lipid composition of the cell. Thus, it is possible that *pex28Δ*, *pex29Δ*, *pex31Δ* and *pex32Δ* mutations result in some lipid changes that, when combined with deletion of *SEI1*, lead to slight changes in LD morphology in the double mutants. However, we believe that those may be peripheral to this study. Instead, here we focus on the dispersed BODIPY structures that are unique to and the dominant phenotype of *sei1Δpex30Δ* cells. We have modified the text to make this point clearer.

3-“Figure 2A: In the text the authors write, “. . .Pex28, Pex29, and Pex32 efficiently co-precipitated with Pex30.” I agree this is true for Pex29, if you compare Pex30 input with Pex30 pulled-down. For the Pex28 and Pex32, only ~20% of Pex30 is pulled down, and it is impossible to know the stoichiometry of Pex30 to the others. So I'd simply omit "efficiently." By the way, as Pex28 mediates the binding of Pex30 to Pex32 (shown later), but Pex32 pulls down Pex30, the authors can further conclude from Fig. 2A that much (most if not all?) of Pex28 is constitutively bound to Pex30.”

We agree with the reviewer. From our analysis, it is impossible to conclude about the relative affinities and binding efficiency. The text has been changed accordingly. In relation, to the second point our data also suggest that Pex28 and Pex32 not bound to Pex30 is unstable.

4-“Figure 3A, 3B: The authors state that "all Pex28 and Pex32 foci were apposed to a peroxisome. . .", but 3B shows the converse, that all peroxisomes were associated with Pex28 (or Pex32) dots. In Fig. 3A, many green dots (in the merge) do not have corresponding red dots.”

We agree with the reviewer. The text has been modified.

5-“Figure S5B: This figure is hard to interpret, as only WT, 2-59Δ and 284-408Δ

are expressed, according to the immunoblot. The pull-downs then show that the amino terminus and the DysF domain are not important for association with the other Pex proteins. But I don't think you can conclude anything about the RHD from this experiment. The fact that significant Pex28 and Pex29 are pulled down by invisible RHD deletions show, if anything, that the intact RHD domain is dispensable for the protein associations, not that "RHD... is critical for Pex30 to bind and stabilize its partners."

We would like to point out that Figure S5B does not correspond to a pull-down/immunoprecipitation. In this experiment, we simply analyse the steady state levels of endogenously tagged Pex28, Pex29 and Pex32 in extracts from cells expressing various Pex30 mutants. We observed that the levels of endogenously tagged Pex28, Pex29 and Pex32 mirror the levels of Pex30. Mutations affecting Pex30 levels (e.g. interfering with RHD) result in destabilization of Pex28, Pex29 and Pex32. A potential source of confusion comes from Pex30(415-513Δ) (also labelled as Pex30DUF4196Δ) since it is not recognized by the anti-Pex30 antibody. This antibody recognises an epitope at the very C-terminus of Pex30 that is absent in this mutant. However, from other experiments we know that this mutant is well expressed and stable (for ex, Figure 1C), resulting in nearly normal levels of Pex28, Pex29 and Pex32. This is now explained in the figure legend.

6-"Figure 5: Along the same lines, the authors swap out RHD from Pex30 with RHDs from other Pex proteins. They don't work for binding partners, leading to their conclusion that, "Pex30 RHD is necessary and sufficient to bind and stabilize Pex28, Pex29, and Pex32." It may be necessary (but see my comment (5) above), but it is only sufficient in the context of the other domains in the other Pex proteins, as these are present in the chimeras. It seems very possible that one needs BOTH Pex30 RHDs AND other elements found downstream in all the PEX proteins of the family. "Sufficient" is too strong a term.

We thank the reviewer for this comment which prompted us to generate a new Pex30 mutant - Pex30(60-283) - lacking both the N-terminus and the C-terminal Dysferlin and DUF4196 domains. When expressed from the endogenous *PEX30* locus this

mutant, which corresponds almost exclusively to the RHD, is stable and binds and stabilizes its partners Pex28, Pex29 and Pex32. This new data is presented in Figures 5A and S5C. Moreover, together with the previous data on the chimeric constructs (now Figure 5B-C), we feel that our conclusion about the necessity and sufficiency of the RHD for the Pex30 interaction with its partners is justified.

Reviewer 3:

1-“In Fig. 1, the authors present data demonstrating that the function of Pex30 in LD biogenesis does not require the other Pex30 family members. They then go on to overexpress the other family members, which are less abundant in cells, to show that when overexpressed, they cannot substitute for Pex30 in LD biogenesis. However, in Fig. 1F, punctate BODIPY-labeled structures can be seen in cells overexpressing Pex29 and to a lesser extent Pex31. Are these structures LD, and, if so, would these results suggest a partial rescue? In addition, the authors should quantify the LD phenotypes observed for the experiments shown in Fig. 1.”

We thank the reviewer for these comments. As stated in response to Reviewer#1 (Point 1) and Reviewer#2 (Points 1 and 2), LD formation in *sei1Δpex30Δ* cells is inhibited but not completely blocked and a few bright foci, corresponding to some aberrant LDs are formed in these cells. We see them in *sei1Δpex30Δ* cells and in cells overexpressing Pex28, Pex29 and Pex31 at similar frequency. More importantly, we observe that the presence of dispersed BODIPY structures, the dominant phenotype of *sei1Δpex30Δ* cells, is unaltered by overexpression of Pex28, Pex29 and Pex31. The text has now been modified and new microscopy images are shown to make this point clearer.

2- “The authors use results obtained with mutants that have partial deletions of the RHD to support their conclusion that the RHD is critical for Pex30 to bind and stabilize its binding partners. However, the Pex30 RHD mutants are either unstable or expressed at levels far lower than WT Pex30. There are no bands visible/extremely faint bands for these proteins in the western blot shown in

Fig. S5. Therefore, it is difficult to conclude, based on these data, that the RHD is critical for Pex30 to bind its partners as the RHD Pex30 mutants may be misfolded or completely destabilized. This needs to be addressed when discussing Fig. S5B.

The chimera data do help support their conclusion regarding the RHD. Is expression of the Pex30 RHD domain alone sufficient to bind and stabilize its binding partners or are the chimeras necessary for proper localization, stability, etc?

The DUF mutant is also not visible in the western blot shown in Fig. S5. It is likely due to the fact that the Pex30 antibody was made against AAs 509-519, which are missing in that mutant. The authors need to make this clear in the legend, or else it is very confusing to the reader.”

We thank the reviewer for this comment and for suggesting to make a RHD only mutant. Please see our response to Reviewer#2, Point 6.

Also, thank you for raising the confusion in relation to the DUF4196 domain mutant. As stated above (Reviewer#2, Point 5) and also pointed out by the reviewer, the anti-Pex30 antibody recognises an epitope at the very C-terminus of Pex30 that is absent in Pex30DUF4196Δ. However, from other experiments we know that this mutant is well expressed and stable, resulting in nearly normal levels of Pex28, Pex29 and Pex32. This is now explained in the figure legend.

3- “In Fig 5E, are the Pex mutants not localizing to the NVJ or is the NVJ not being formed/delayed formation? For Fig. 5E, the legend describes that the cells shown are also labeled with NVJ and LD markers, however only Pex protein localization is shown. The authors need to show the NVJ and LD markers in the figure along with the Pex protein localization. In addition, the authors need to add quantification for the percentage of cells in which NVJs are visible as well as quantification of Pex protein localization to the NVJ. For 5F, a better description of the quantification is needed. Does the quantification only include cells that have an NVJ or are all cells included? Three categories should be quantified: no

NVJ, NVJ with no LD cluster, NVJ with LD cluster. A similar analysis should also be done for Fig. 4E.”

We thank the reviewer for the suggestion regarding the NVJ phenotypes. These have now been re-analysed in cells expressing fluorescently tagged Nvj1 so that both the NVJ and the LDs (labelled with MDH) could be visualized (now Figure 5F). We also repeated the quantifications including the three categories described (now Figure 5E). These new experiments confirmed that Pex31^{Pex30RHD} and Pex30^{DysFA} have a LD clustering defect indistinguishable from *pex30Δ* cells.

In addition, the legend of Figure 4E now includes a description of how the quantification was done.

Minor comments:

1- “It would be very useful to better describe the LD phenotype for the *sei1* mutant in the results section for readers less familiar with the field. It would make it easier to interpret the LD images in Fig. 1.”

We agree with the reviewer and now the text includes a more complete description of the LD phenotype in *sei1Δ* and *sei1Δpex30Δ* cells.

2-“For the figures in which stationary phase cells are shown, it would be very helpful to the reader to indicate within the figure itself which panels are showing stationary phase cells - for example, 3D, 4A, and 5E. It is stated in the legend but would also be useful if indicated in the figure.”

We thank the reviewer for this suggestion. The growth phase information is now included in Figures 3, 4 and 5.

3- “The authors use Fig. 4A to conclude that Pex30 fails to localize to the NVJ in a *pex29* mutant. They provide clear evidence of this in Fig 4B, but Fig. 4A cannot be used to make such a definitive statement as they are not looking at an NVJ marker. The wording regarding the conclusion to Fig. 4A needs to be modified. Also, in Fig. 4D, what is meant by Nvj1 clusters? Is it localization of Nvj1 to an NVJ? –The word cluster here is confusing.”

We agree with the reviewer. Since we did not include a NVJ marker in this experiment, the absence of Pex30 brighter structures during stationary phase observed in *pex29Δ* cells only suggests that Pex30-mNG fails to localize to the NVJ. We have changed the text to accommodate this limitation.

We again agree with the reviewer that the description “Nvj1 clusters” is confusing. The structure labelled as a “cluster” is indeed the NVJ. The labelling has now been corrected.

4-“The legend for Fig. 4C refers to quantifications in 4B but there are no quantifications in 4B.”

This has been fixed.

5-“In the results section that describes Fig. 5 (end of page 14), panels C and D are misreferenced.”

This has been fixed.

July 13, 2021

RE: JCB Manuscript #202103176R

Dr. Pedro Carvalho
University of Oxford
Sir William Dunn School of Pathology
South Parks road
South Parks Road
Oxford, UK OX1 3RE
United Kingdom

Dear Dr. Carvalho:

Thank you for submitting your revised manuscript entitled "Pex30-like proteins function as Pex30 adaptors at distinct ER membrane contact-sites". We would be happy to publish your paper in JCB pending final revisions necessary to meet our formatting guidelines (see details below). In your final revision, please be sure to address reviewer #3's remaining comments.

A. MANUSCRIPT ORGANIZATION AND FORMATTING:

Full guidelines are available on our Instructions for Authors page, <https://jcb.rupress.org/submission-guidelines#revised>. **Submission of a paper that does not conform to JCB guidelines will delay the acceptance of your manuscript.**

1) Text limits: Character count for Articles is < 40,000, not including spaces. Count includes title page, abstract, introduction, results, discussion, acknowledgments, and figure legends. Count does not include materials and methods, references, tables, or supplemental legends.

2) Figures limits: Articles may have up to 10 main text figures.

3) Figure formatting: Scale bars must be present on all microscopy images, including inset magnifications. Molecular weight or nucleic acid size markers must be included on all gel electrophoresis.

4) Statistical analysis: Error bars on graphic representations of numerical data must be clearly described in the figure legend. The number of independent data points (n) represented in a graph must be indicated in the legend. Statistical methods should be explained in full in the materials and methods. For figures presenting pooled data the statistical measure should be defined in the figure legends. Please also be sure to indicate the statistical tests used in each of your experiments (either in the figure legend itself or in a separate methods section) as well as the parameters of the test (for example, if you ran a t-test, please indicate if it was one- or two-sided, etc.). Also, if you used parametric tests, please indicate if the data distribution was tested for normality (and if so, how). If not, you must state something to the effect that "Data distribution was assumed to be

normal but this was not formally tested."

5) Abstract and title: The abstract should be no longer than 160 words and should communicate the significance of the paper for a general audience. The title should be less than 100 characters including spaces. Make the title concise but accessible to a general readership.

* We suggest a slightly abbreviated version of your title: Pex30-like proteins function as adaptors at distinct ER membrane contact-sites

6) Materials and methods: Should be comprehensive and not simply reference a previous publication for details on how an experiment was performed. Please provide full descriptions in the text for readers who may not have access to referenced manuscripts.

7) Please be sure to provide the sequences for all of your primers/oligos and RNAi constructs in the materials and methods. You must also indicate in the methods the source, species, and catalog numbers (where appropriate) for all of your antibodies. Please also indicate the acquisition and quantification methods for immunoblotting/western blots.

8) Microscope image acquisition: The following information must be provided about the acquisition and processing of images:

a. Make and model of microscope

b. Type, magnification, and numerical aperture of the objective lenses

c. Temperature

d. Imaging medium

e. Fluorochromes

f. Camera make and model

g. Acquisition software

h. Any software used for image processing subsequent to data acquisition. Please include details and types of operations involved (e.g., type of deconvolution, 3D reconstitutions, surface or volume rendering, gamma adjustments, etc.).

10) Supplemental materials: There are strict limits on the allowable amount of supplemental data. Articles/Tools may have up to 5 supplemental display items (figures and tables). Please also note that tables, like figures, should be provided as individual, editable files. A summary of all supplemental material should appear at the end of the Materials and methods section.

13) ORCID IDs: ORCID IDs are unique identifiers allowing researchers to create a record of their various scholarly contributions in a single place. At resubmission of your final files, please consider providing an ORCID ID for as many contributing authors as possible.

B. FINAL FILES:

Thank you for this interesting contribution, we look forward to publishing your paper in Journal of Cell Biology.

Sincerely,

Jodi Nunnari, Ph.D.
Editor-in-Chief
Journal of Cell Biology

Reviewer #2 (Comments to the Authors (Required)):

Please see my original review regarding summary, advance, etc. In brief, the manuscript expands our view of the function of the Pex30 family in organelle contacts and LD assembly.

The revision addresses all my concerns, all of which dealt with interpretation of results. I have no new concerns.

Reviewer #3 (Comments to the Authors (Required)):

The authors address the majority of the concerns raised, and the revised manuscript is stronger and clearer with the additions and modifications. As mentioned in the first round of review, the characterization of the two distinct Pex30 complexes at both the biochemical and cell biological levels is well done and the findings are exciting.

There are a few minor points, that if addressed, would help the reader better understand the study.

For the LD phenotypes in Fig. 1, the authors do a better job explaining the phenotypes observed in the revision, but the results could still benefit from a slightly more detailed description of the data. For example, in the response to reviewers the authors state "LD formation in *sei1Δpex30Δ* cells is inhibited but not completely blocked and a few bright foci, corresponding to some aberrant LDs are formed in these cells. We see them in *sei1Δpex30Δ* cells and in cells overexpressing Pex28, Pex29 and Pex31 at similar frequency." It might be beneficial to add some version of the text used to respond to reviewers to the results section as the responses helped to further clarify the data and interpretations for me.

The addition of the Pex30(60-283) truncation and the resulting data nicely help make the point for the sufficiency of the RHD. For clarity, it might be useful to include the schematic of that truncation in Fig. 5, not just in the supplement, or at least label the schematic of Pex30 in Fig. 5 with amino acid numbers. This would provide the reader with a better visual of what parts of Pex30 are included in aa 60-283, which would help highlight the significance of the data shown in Fig. 5A.

The additional quantification shown in Fig. 5G is very useful. However, a better description and interpretation of these data and the phenotypes observed would really benefit the reader. Similar to a *pex30* deletion, the authors are seeing a delay/inhibition of NVJ formation in the Pex30 mutants as well as a LD clustering defect. However, the delay/inhibition in NVJ formation is not discussed in this section of the results.